# Immediate nuclear accumulation of BMAL1 to regulate cellular circadian clock synchronization
Teruya Tamaru [1,6] ✉, Genki Kawamura[2,6], Hikari Yoshitane [3,4], Satoshi Koinuma[5], Yoshitaka Fukada [3,4], Atsuhiko Naito [1], Takeaki Ozawa [2] ✉ & Ken Takamatsu[1]

Cell-autonomous circadian clocks coordinate daily physiological timing, allowing them to synchronize with the environment. However, the initial signaling events shared by diverse synchronizing cues remain elusive. Here, we show that changes in the clock protein localization serve as a common synchronizing event by investigating the relationship between BMAL1 and CLOCK localization patterns and clock synchronization in NIH-3T3 fibroblasts. We demonstrate synchronized nuclear BMAL1 accumulation as an immediate synchronization response (ISR), as well as CLOCK accumulation following various clock-resetting treatments. BMAL1-Ser90 phosphorylation by CK2, which is reported to promote nuclear BMAL1 accumulation, is also immediately elevated. Importantly, pharmacological CK2 inhibition partially suppresses the acute *Per2* increase and clock reset. Furthermore, computational simulation supports that an increase in the BMAL1 phosphorylation levels and its subsequent nuclear localization could reset the clock. In summary, our findings suggest that BMAL1-ISR is a key event that acts as an integrative switching signal to link molecular clock oscillation and diverse synchronization cues.

The circadian clock is an autonomous cellular timing system that oscillates with a 24-h periodicity[1,2]. This system regulates the global gene expression and metabolism in nearly every cell throughout the entire body of living organisms[3,4]. A distinctive feature of the circadian clock is its ability to synchronize the circadian phase of each cell in response to external factors, including growth factors and stress stimulations[5]. This cellular synchronization process contributes to the coordination of homeostasis and environmental adaptation at the cellular, tissue, and individual levels[6].

The impetus for this study was our paper published in 2003[7]. In that paper, we found that the nucleocytoplasmic BMAL1 localization in serum-starved mouse fibroblast NIH-3T3 cells cultured for several days without clock synchronization become heterogeneous compared to that of the early time after synchronization. Specifically, we observed apparently homogenous rhythmic cellular distribution pattern of cellular BMAL1 localization, where BMAL1 was predominantly in the nucleus at 4 h post clock synchronization, followed by an enrichment in the cytoplasm at 16 h, and again in the nucleus at 24–28 h. The nuclear entry of cytoplasmic BMAL1 and its interaction with CLOCK, leading to the assembly of a

BMAL1:CLOCK heterodimer, likely transactivates *Per1/2* clock gene expression[7–9]. An acute *Per1/2* surge is considered a key event following clock synchronization[10,11]. Therefore, homogenous BMAL1 accumulation in the nucleus could potentially synchronize the circadian clock.

Circadian clocks have diverse intracellular pathways for responding to different environments and stresses[5,12,13], however, it remains unclear what kind of regulatory signals clock proteins carry when cellular clocks are synchronized and phase-shifted by different clock synchronization factors. More specifically, critical intracellular signal that triggers BMAL1 nuclear accumulation and its relationships with clock synchronization remain unclear. Thus, we investigated the role of immediate change in nuclear BMAL1 and CLOCK accumulation, as a common initial clock protein signal switch during clock synchronization by various external cues.

BMAL1 contains potential nuclear localization and export signals, and the phosphorylation status near these localization signals controls the subcellular localization of BMAL1 and its dimerization partner CLOCK[8,14]. Among the post-translational modifications of BMAL1, CK2-mediated BMAL1 phosphorylation at Ser90 shows a circadian rhythm, and inhibition

[1]Department of Physiology & Advanced Research Center for Medical Science, Toho University School of Medicine, Ohta-ku, Tokyo, Japan. [2]Department of Chemistry, School of Science, The University of Tokyo, Bunkyo-ku, Tokyo, Japan. [3]Circadian Clock Project, Tokyo Metropolitan Institute of Medical Science, Setagaya-ku, Tokyo, Japan. [4]Department of Biological Sciences, School of Science, The University of Tokyo, Bunkyo-ku, Tokyo, Japan. [5]Department of Anatomy and Neurobiology, Kindai University Faculty of Medicine, Sakai-City, Osaka, Japan. [6]These authors contributed equally: Teruya Tamaru, Genki Kawamura. ✉e-mail: tetamaru@med.toho-u.ac.jp; ozawa@chem.s.u-tokyo.ac.jp

of BMAL1-Ser90 phosphorylation reduces nuclear BMAL1 accumulation[14]. Accordingly, the nucleocytoplasmic BMAL1 localization may reflect a desynchronized state of the cellular clock phase. In addition, BMAL1-Ser90 phosphorylation is essential for circadian *Per2* expression and is a prerequisite event for circadian feedback repression by CRY of clock gene expression through BMAL1-acetylation[15]. However, there has been a lack of evidence concerning the mechanisms that regulate BMAL1, particularly regarding the subcellular localization change of BMAL1 during the early response to clock synchronizers. Therefore, we hypothesized that the immediate synchronization response (ISR) of BMAL1, a synchronous accumulation of BMAL1 and CLOCK in the nucleus immediately after cellular clock synchronization mediated by BMAL1 phosphorylation, is a clock protein signal that acts as one of the switch on the synchronous oscillation of cellular clocks.

In this study, we used mouse fibroblasts as a cellular clock model and investigated the role of BMAL1-ISR in circadian clock synchronization by cellular and molecular biological analyses and numerical simulations. First, we examined the change in nucleocytoplasmic BMAL1 localization upon treatment with synchronizing factors and found that BMAL1 accumulation after stimulation was pronounced at circadian times when BMAL1 localization was more cytoplasmic. Second, we analyzed the effect of synchronizing factors on BMAL1 phosphorylation and found that BMAL1 phosphorylation coincides with BMAL1 nuclear entry. Finally, we evaluated BMAL1 phosphorylation as a key event for facilitating clock synchronization and maintaining the circadian oscillation.

## Results

### Immediate change in nucleocytoplasmic BMAL1 and CLOCK localization pattern after clock-synchronization

Based on the hypothesis that the nucleocytoplasmic localization of BMAL1 and CLOCK reflects circadian multicellular clock synchronization, we investigated the localization of BMAL1 and CLOCK in a cellular clock model, NIH-3T3 fibroblasts, by confocal imaging of immunocytochemically stained samples during the early stage of clock synchronization. In unsynchronized NIH-3T3 cells (NIH-3T3 WT) cultured for several days, as expected, the localization of the nucleocytoplasmic BMAL1 signal varied from cell to cell and showed a slightly nuclear-shifted localization as did CLOCK (Fig. 1a). Intracellular distribution of CLOCK was also observed in NIH-3T3 cells depleted of BMAL1 by genome editing (NIH-3T3 BMAL1-KO), which showed attenuated nuclear CLOCK signal compared to NIH-3T3 WT cells (Fig. 1a). However, after clock-synchronizing stimulation with dexamethasone (Dex), BMAL1 rapidly accumulated into the nucleus (Nuc/Cyto ratio; more than 2) in most cells at 20 min (Dex 20 min) and at 40 min (Dex 40 min) post-stimulation. The accumulation of CLOCK signal in the nucleus also appeared more pronounced at Dex 40 min than at Dex 20 min. Quantitative image analysis of BMAL1 and CLOCK localization using the Fiji-based method (Supplementary Fig. 1) revealed that the nucleocytoplasmic ratio of BMAL1 signal increased from Dex 20 min to Dex 40 min, as shown by the distribution change in the histograms (Fig. 1b). This difference was significant between no treatment (control) and Dex 20 min or Dex 40 min, respectively (Fig. 1c). CLOCK nuclear translocation followed BMAL1, with a significant increase in the CLOCK nuclear-to-cytoplasmic ratio observed only at Dex 40 min (Fig. 1b, c). The shift in the distribution of nucleocytoplasmic CLOCK localization was further confirmed with another lot of antibodies (CLOCKcst) (Supplementary Fig. 2). Notably, quantification by double immunofluorescence staining of BMAL1 and CLOCK revealed a positive correlation between the nucleocytoplasmic ratios of the two proteins. Interestingly, the correlation coefficient decreased transiently at Dex 20 min but increased at Dex 40 min compared to the control, suggesting that BMAL1:CLOCK dimerization is more likely to occur at Dex 40 min than for Dex 20 min (Fig. 1d). Since CLOCK nuclear localization is dependent on BMAL1[8,16], increase in the correlation over time suggest clock proteins translocate to the nucleus sequentially in the order of BMAL1 and CLOCK. These results demonstrate that ISR in which clock proteins accumulate in the nucleus sequentially in the order of BMAL1 and CLOCK following clock-synchronizing stimuli.

Next, to investigate the commonality of the ISR among cell types, we used confocal imaging to examine whether the rapid accumulation of BMAL1 and CLOCK in the nucleus occurs after Dex stimulation in rodent cells with clock oscillation including rat glioma C6[17], mouse myoblast C2C12[18], and mouse fibroblast MEF cells[14] (Supplementary Fig. 3). C6 cells showed a slightly cytoplasmic enriched distribution of BMAL1, whereas BMAL1 localization was slightly nuclear for C2C12 and MEF cells. In all cells, BMAL1 nuclear translocation was observed, and CLOCK also tended to be enriched in the nucleus after the stimulation for MEF cells (Supplementary Fig. 3b). These results suggest that BMAL1-ISR is preserved among diverse types of cells with a clock function. In addition, to test whether BMAL1-ISR occurs, we evaluated stimulation-mediated nuclear accumulation of BMAL1 with various clock synchronizers: mitogen (EGF), cAMP activator (Forskolin), calcium ionophore (A23187), and cellular stress (Heat-shock). In all cases, a homogenous accumulation of BMAL1 signal in the nuclei of NIH-3T3 cells was observed 20 min after each stimulation (Supplementary Fig. 4a). The distribution of quantified nuclear-to-cytoplasmic ratio also supported that there was an increase in the ratio for BMAL1 upon all the stimulation tested (Supplementary Fig. 4b, c). CLOCK accumulated uniformly in the nuclei 20 min after stimulation, as in the case of Dex stimulation, to a lesser extent compared to BMAL1. This result indicates that the BMAL1-ISR, a putative trigger of clock synchronization, was shown to be caused by various clock synchronizers.

### Circadian time-dependent nucleocytoplasmic localization pattern changes of BMAL1 and CLOCK

We have previously shown a circadian fluctuation in BMAL1 intracellular distribution in the synchronized cells[7]. Therefore, we investigated whether the acute response of BMAL1 and CLOCK localization change also occurs during the phase shift of cellular clocks after clock synchronization in a circadian time-dependent manner. For this purpose, we examined BMAL1 and CLOCK abundance in the nuclear and cytoplasmic compartments of cells that were presynchronized with Dex for 15, 18, and 22 h before a second synchronization stimulation.

At 15 and 18 h after Dex synchronization (Dex 15 h and Dex 18 h), BMAL1 and CLOCK were predominantly cytoplasmic (Fig. 2a and Supplementary Fig. 5a). Following a second Dex stimulation, there was a significant increase in the nuclear-to-cytoplasmic ratio for both BMAL1 and CLOCK at Dex 20 min and Dex 40 min compared to the unstimulated control (Fig. 2b, c and Supplementary Fig. 5b). Notably, no significant differences in the nucleocytoplasmic ratio of stained BMAL1 and CLOCK signals were observed between Dex 20 min and Dex 40 min at either time point, suggesting that the peak of circadian nuclear enrichment of BMAL1 and CLOCK occurred around 20 min after the stimulation (Fig. 2c). In contrast, at 22 h post-Dex synchronization (Dex 22 h), BMAL1 and CLOCK were predominantly nuclear, and the nucleocytoplasmic ratio at Dex 22 h was significantly higher compared to Dex 15 h or Dex 18 h (Fig. 2a, b). The broader distribution of BMAL1 and CLOCK nucleocytoplasmic ratio at Dex 22 h (Supplementary Fig. 5b) may be attributed to the observation that many cells exhibit a very weak signal in the cytoplasm. Moreover, no significant differences in the nucleocytoplasmic ratio distribution of BMAL1 and CLOCK were observed among samples at Dex 22 h (Fig. 2c). Using the nuclear/cytosolic fractionation approach, we further confirmed that nuclear BMAL1 abundance increased at 14 and 18 h, but not at 22 h post Dex synchronization (Fig. 2d, e). These results collectively demonstrate that clock-synchronizing stimulation at Dex 15 h and Dex 18 h induces ISR, whereas at Dex 22 h, BMAL1:CLOCK was already enriched in the nucleus and no ISR was observed.

These findings suggest that the ISR occurs in a circadian time-dependent manner in synchronized cells. Specifically, it occurs more synchronously when BMAL1:CLOCK is relatively cytoplasmic, indicating a greater responsiveness when the heterodimer is readily available for nuclear translocation.

**Fig. 1 | Immediate change in nucleocytoplasmic BMAL1:CLOCK pattern following clock-synchronization of NIH-3T3 fibroblast by Dex treatments. a** Representative immunofluorescence images of BMAL1 and CLOCK showing immediate nuclear accumulation of BMAL1 and then CLOCK. NIH-3T3 (WT) and BMAL1-deficient (BMAL1-KO expressing Bmal1-Luc) cells were treated with 100 nM dexamethasone (Dex) for 20–40 min for clock synchronization or left untreated ("Unstimulated", negative control). Cells were fixed and stained with anti-BMAL1 (rabbit polyclonal Nt; green) and anti-CLOCK (mouse monoclonal CLSP3; red) antibodies, followed by DAPI (nuclear staining; cyan) and visualized by confocal imaging. Scale bars: 10 μm. **b**, **c** Quantification of nuclear and cytoplasmic CLOCK levels using a custom Fiji-based analysis procedure as described in Supplementary Fig. 1. **d** Scatter plots and calculated correlation coefficients between the nuclear/cytoplasmic ratios of BMAL1 and CLOCK. Control: $n = 5$ independent observation with 103 cells, Dex 20 min: $n = 5$ independent observation with 96 cells, Dex 40 min: $n = 5$ independent observation with 101 cells. ***$p < 0.001$, not significant unless mentioned, two-tailed Welch's $t$-test.

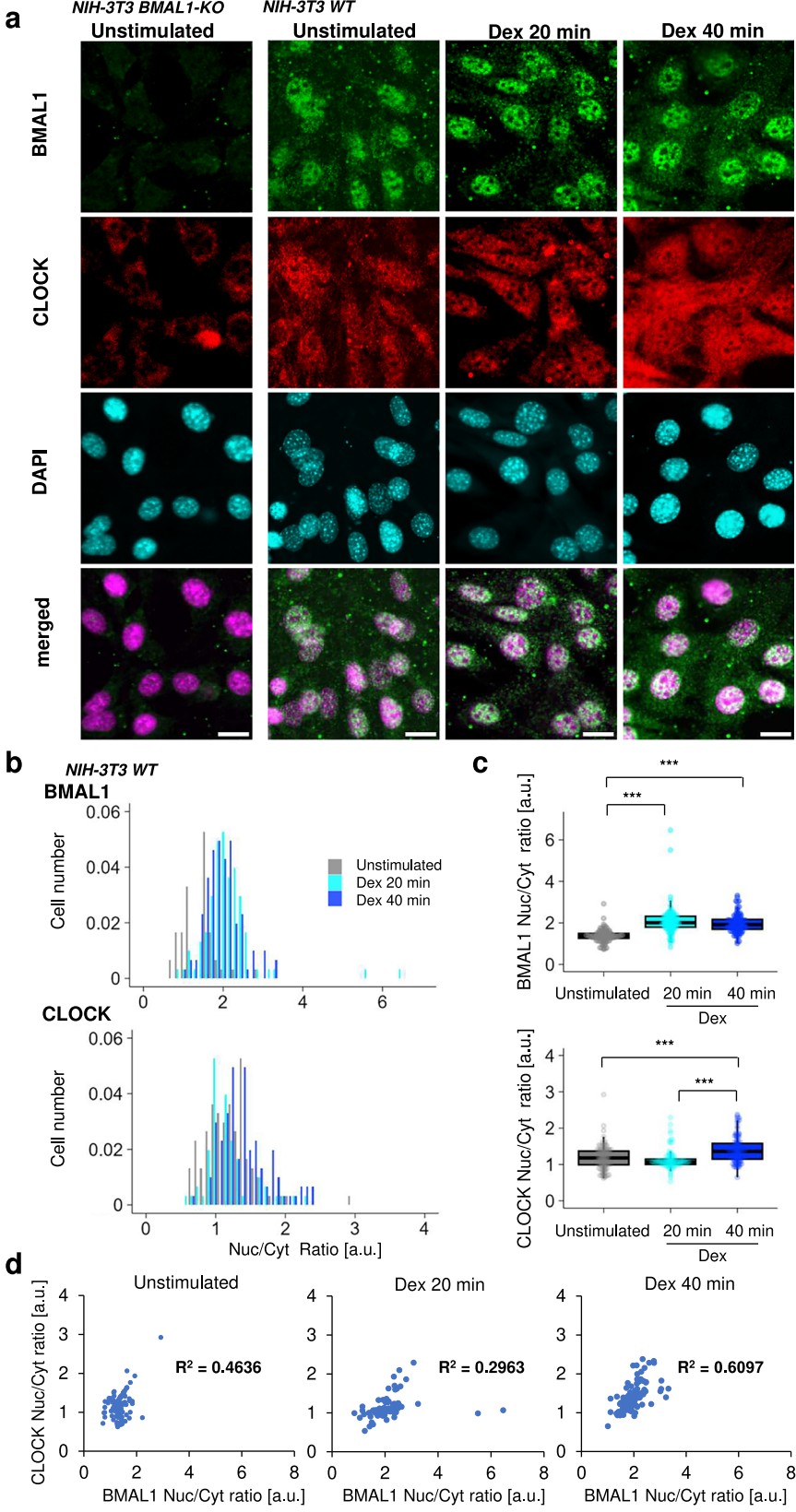

## Single-cell analysis of BMAL1-ISR through live-cell imaging

To further investigate the temporal dynamics of the BMAL1-ISR phenomenon within the same single-cell, we conducted live-cell imaging of BMAL1 translocation following clock-synchronizing stimulation. To achieve this, we generated a fusion protein of BMAL1 and a fluorescent protein, monomeric Venus (mVenus), and introduced it into BMAL1-deficient NIH-3T3 (NIH-3T3 BMAL1-KO). We introduced a point mutation into the original Venus (A206K) to prevent the fusion protein from forming an unphysiological dimer that could affect the localization[19]. Furthermore, we evaluated two types of peptide linkers connecting BMAL1

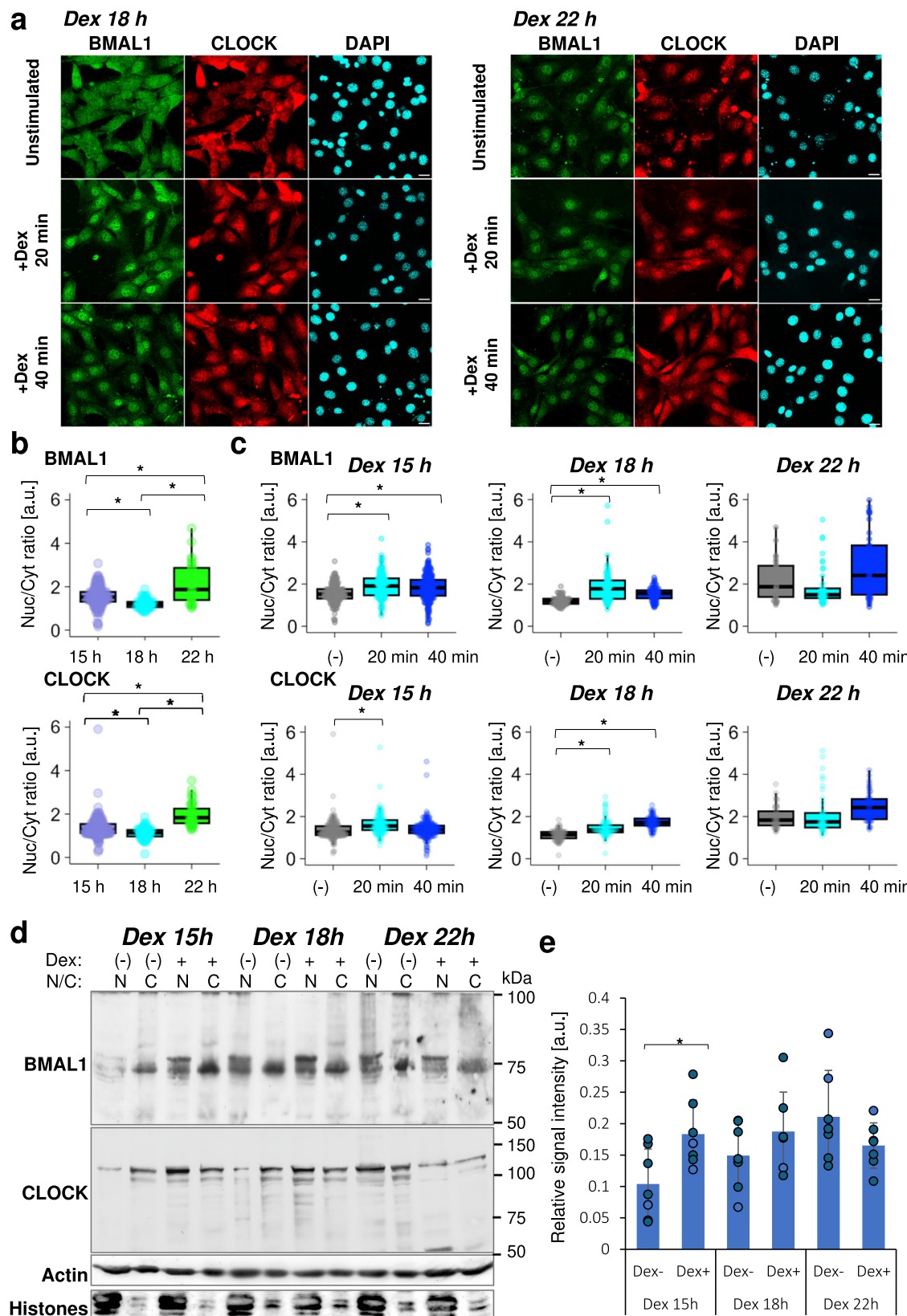

and mVenus (mVenus-linker-BMAL1, Supplementary Fig. 6a). We confirmed the circadian functionality of these reporters by demonstrating their ability to rescue the circadian Bmal1-Luc oscillation in BMAL1-KO cells (average period ± SD of three experiments; linker A: 24.4 ± 2.1 h, linker B: 24.4 ± 1.5 h; Supplementary Fig. 6b). For the localization of BMAL1, we

found that linker A type reporter showed a heterogeneous nucleocyto-plasmic BMAL1 distribution among cells in the asynchronous state mirroring the localization pattern of endogenous BMAL1 (Fig. 1a and Supplementary Fig. 6c). In contrast, the linker B-type reporter showed a nuclear-enriched localization with aggregation in the cytoplasm

**Fig. 2 | Circadian time-dependent immediate modulation of the nucleocytoplasmic BMAL1:CLOCK pattern in response to clock-synchronizing treatment.**
**a** Representative immunofluorescence images of BMAL1 and CLOCK showing a circadian time-dependent immediate nuclear accumulation of BMAL1 and CLOCK. NIH-3T3 cells were initially treated with Dex for clock synchronization. At 15, 18 or 22 h post Dex-mediated synchronization (Dex 15 h, Dex 18 h, and Dex 22 h), cells were either re-treated with Dex for 20–40 min to reset the clock or left unstimulated as a negative control ("Unstimulated"). Cells were fixed and stained with anti-BMAL1 and anti-CLOCK antibodies, followed by DAPI, and visualized by confocal imaging. The upper panels show representative images from Dex 18 h samples and the lower panels show representative images from Dex 22 h samples. Representative images from Dex 15 h samples are shown in Supplementary Fig. 5a. Scale bar: 10 μm. **b**, **c** Quantification of nuclear and cytoplasmic BMAL1:CLOCK levels using custom Fiji-based analysis. **b** Box-and-whisker plot of the nuclear-to-cytoplasmic ratio of BMAL1 and CLOCK in unstimulated cells. Data corresponds to "Unstimulated" samples in (**a**). **c** Box-and-whisker plot of the nuclear-to-cytoplasmic ratio of BMAL1 and CLOCK after Dexamethasone stimulation for 20 and 40 min. "(−)"

denotes unstimulated samples. Dex 15 h: Unstimulated: $n = 4$ independent observation with 183 cells, Dex 20 min: $n = 4$ independent observation with 132 cells, Dex 40 min: $n = 4$ independent observation with 260 cells; Dex 18 h: Unstimulated: $n = 4$ independent observation with 83 cells, Dex 20 min: $n = 6$ independent observation with 82 cells, Dex 40 min: $n = 4$ independent observation with 77 cells; Dex 22 h: Unstimulated: $n = 4$ independent observation with 40 cells, Dex 20 min: $n = 4$ independent observation with 47 cells, Dex 40 min: $n = 3$ independent observation with 49 cells. $*p < 0.05$, not significant unless mentioned, two-tailed Welch's $t$-test. **d** Detection of BMAL1 and CLOCK protein abundance in the nucleus and cytosol after Dex stimulation. Cells were synchronized with Dex and harvested at CT 15, 18, and 22 h. Subcellular fractionation was performed to separate nuclear and cytosolic fractions. Western blot analysis was conducted to quantify BMAL1 and CLOCK protein levels in each fraction. "N" denotes nuclear fraction and "C" denotes cytosolic fraction. Actin and histone were used as loading controls for cytosolic and nuclear fractions, respectively. **e** Quantification of nuclear BMAL1 level from band intensities of Western blot. $n = 7$ independent experiments, $*p < 0.05$.

---

(Supplementary Fig. 6c). Therefore, we selected the linker A-type reporter for subsequent experiments to visualize BMAL1.

Using this BMAL1 reporter, we observed time-lapse changes in BMAL1 localization after Dex and EGF stimulation as clock synchronizers. First, we confirmed that both Dex and EGF stimulation synchronized the circadian clock in the reporter-expressing cell line by monitoring Bmal1-Luc activity (Fig. 3a and Supplementary Fig. 6d). Then, we conducted time-lapse imaging of the mVenus-linker-BMAL1 reporter upon Dex and EGF stimulation (Fig. 3b and Supplementary Movie 1–3). We found the nuclear BMAL1 fluorescence intensity increased following both stimulations, indicating that BMAL1 translocated to the nucleus in response to the stimulation. To quantitatively assess the temporal change in BMAL1 localization, we established a pipeline to track the change in nuclear BMAL1 localization levels by harnessing cellular segmentation and single-cell tracking, employing H2B-mKate2 as a nucleus marker (Supplementary Fig. 7a). Due to the limitation in the live cell imaging setup that was designed to minimize cellular damage from the excitation light, the cytoplasmic fluorescence intensity of the BMAL1 reporter was undetectable in some cells. For this reason, we used the ratio of nuclear to total cellular fluorescence intensity as a measure of BMAL1 nuclear localization in this analysis (Supplementary Fig. 7b). We found that BMAL1 nuclear levels increased within 30 min after clock-synchronizing stimulation, reaching a peak at approximately 0.5–1 h of Dex and EGF treatment, with a slightly more pronounced response for Dex (Fig. 3c). Notably, the BMAL1 nuclear ratio exhibited a shift towards higher values after both Dex and EGF treatment, as visualized by the histograms (Fig. 3d).

To further confirm that these BMAL1 nuclear translocations are associated with clock synchronization, we introduced analytical approaches to assess the positive rate and correlation of BMAL1 localization changes among cells. First, to examine the percentage of cells that underwent nuclear translocation of BMAL1 after stimulation, we calculated the positive rate of BMAL1 nuclear localization by analyzing the change in the nuclear-to-cytoplasmic ratio of each cell. When taking the 75% quartile of the ratio for the unstimulated control as a threshold, the results showed that 74% of cells underwent nuclear translocation with EGF, and 80% with Dex stimulation, respectively (Supplementary Fig. 7c). Next, we examined the correlation of BMAL1 localization changes among neighboring cells (local correlation index) and across the entire cell population (global correlation index) (Fig. 3e and Supplementary Fig. 7d). We found a significant increase in the local correlation index upon Dex and EGF stimulation, suggesting that both stimuli induce coordinated nuclear translocation of BMAL1 among neighboring cells (Fig. 3f). In addition, a comparison of local and global indices indicated that correlated BMAL1 localization changes were not confined to specific cell populations but rather occurred throughout the entire stimulated cell population. We also evaluated the time evolution of these correlation indices and found that the correlation of localization increased after the stimulation (Supplementary Fig. 7e). Since both Dex and

EGF synchronize the multicellular circadian clock (Fig. 3a), these findings suggest that nuclear BMAL1 translocation is a response that serves as an early intracellular event that triggers clock synchronization.

## BMAL1-Ser90 phosphorylation by CK2 mediates BMAL1 nuclear translocation

Next, we sought to identify the molecular mechanism that mediates BMAL1 nuclear accumulation. The shuttling of BMAL1 between the nucleus and cytoplasm is regulated by its post-translational modification. Our previous studies have shown that CK2 phosphorylates BMAL1 at Ser90 in a circadian manner. In addition, the mutation replaces Ser90 in BMAL1 with Ala, which prevents phosphorylation by CK2 and thus mimics a constitutively unphosphorylated state (BMAL1-S90A mutation), suppresses BMAL1 nuclear accumulation and impairs the circadian rhythm[14,15]. Therefore, we hypothesized that Ser90 phosphorylation is a critical event for BMAL1 nuclear entry following stimulation with synchronizing factors.

Indeed, we observed a rapid increase in the phosphorylation level of BMAL1-Ser90 within 15 min of post Dex and EGF stimulation (Fig. 4a and Supplementary Fig. 8a). Importantly, this increase in the phosphorylation level was inhibited when cells were co-treated with the CK2 inhibitor, GO289 (Fig. 4b and Supplementary Fig. 8b)[20]. Moreover, the BMAL1-Ser90 phosphorylation signal was also increased in nuclear fractionated samples by Dex stimulation but suppressed by GO289 stimulation, supporting that CK2 inhibition suppresses nuclear BMAL1 phosphorylation at Ser90 (Supplementary Fig. 8c). In contrast, we did not observe an increase in BMAL1 phosphorylation at Ser90 in cells re-stimulated with Dex at Dex 22 h, a time point when BMAL1 predominantly localized in the nucleus (Fig. 4a, b). To further evaluate the impact of CK2 inhibition on BMAL1 nuclear translocation, we examined the effect of GO289 on the BMAL1 localization pattern following synchronizing factor stimulation. We found that GO289 treatment significantly suppressed BMAL1 nuclear accumulation in cells re-stimulated with Dex at Dex 15 h (Supplementary Fig. 8d). These findings suggest that Dex and EGF stimulation induces an acute BMAL1-Ser90 phosphorylation surge by CK2, presumably in a circadian time-dependent manner, driving nuclear BMAL1 accumulation.

To assess the functional significance of CK2-mediated BMAL1 phosphorylation on circadian clock synchronization, we analyzed the effect of transient CK2 inhibition on the circadian Per2-Luc oscillation in NIH-3T3 cells[12]. We first synchronized cells with Dex and then resynchronized them 15 h later, either in the presence or absence of GO289 (Fig. 4c). We observed an increase in the amplitude of the Per2-Luc oscillation by Dex was abolished with the treatment of GO289, indicating decrease in the synchrony among multicellular circadian clock oscillations (Fig. 4d). Furthermore, transient GO289 stimulation suppressed the phase shift induced by Dex, suggesting that CK2 activity is essential for the clock synchronization (Fig. 4d). These findings were further corroborated by the suppression of Dex-induced phase shifts in Bmal1-Luc expressing cells upon CK2

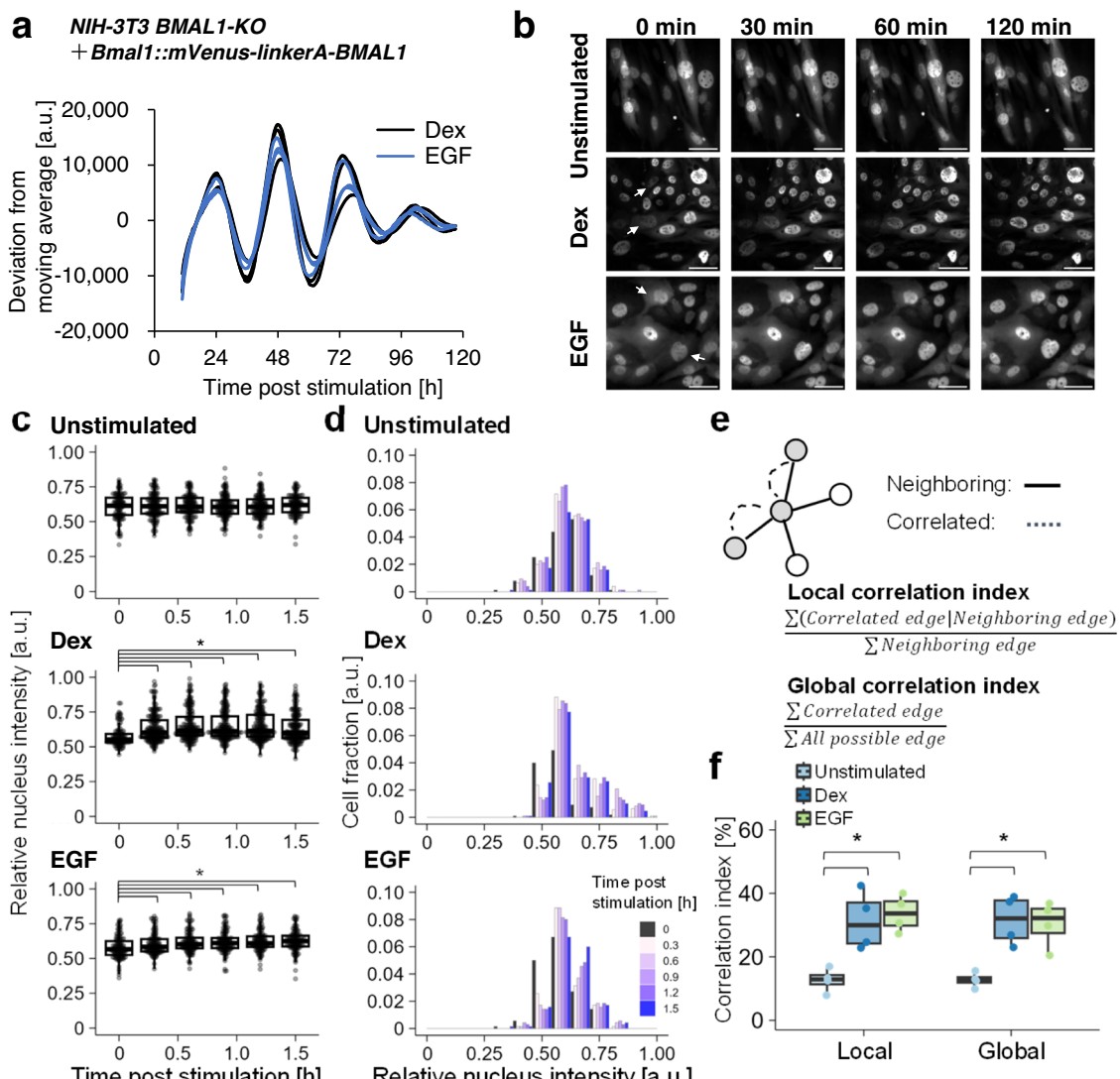

**Fig. 3 | Single-cell analysis of BMAL1 localization dynamics in response to clock-synchronization. a** NIH-3T3 BMAL1-deficient (BMAL1-KO) cells harboring the Bmal1-Luc reporter were stimulated by Dex (100 nM) or EGF (100 ng mL$^{-1}$), and the temporal variation of Bmal1-Luc expression was monitored by a bioluminescence assay. Representative traces after subtracting the deviation from the moving average of the raw data are shown for three experiments. **b** Representative images of temporal BMAL1 localization change upon Dex (100 nM) and EGF (100 ng mL$^{-1}$) stimulation for clock-synchronization or unstimulated at 0-, 30-, 60-, and 120-min post-stimulation. White arrows indicate representative cells that have undergone nuclear accumulation. Scale bar: 50 μm. Images correspond to: Unstimulated: Supplementary Movie 1, Dex: Supplementary Movie 2, EGF: Supplementary Movie 3, respectively. **c** Box plot representation of the temporal localization changes of mVenus-BMAL1 reporter following Dex or EGF stimulation, or unstimulated.

Unstimulated: $n = 4$ independent observation with 93 cells, Dex: $n = 4$ independent observation with 135 cells, EGF: $n = 4$ independent observation with 63 cells. Each dot represents the data from each cell. Relative nuclear intensity is calculated as the ratio of nuclear fluorescence intensity to total fluorescence intensity for the whole cell area. *$p < 0.05$, not significant unless mentioned, two-tailed Welch's $t$-test. **d** A histogram representation of temporal localization changes of BMAL1 reporter after Dex stimulation, EGF stimulation, or unstimulated. The distribution of nuclear fluorescent intensities for each timepoints were represented according to the color-code in the figure legend. **e** Quantification of local and global cell-to-cell correlations in BMAL1 nuclear translocation dynamics. The local and global correlation indices were calculated as described in the "Material and methods" section. **f** Box plot representation of the local and global correlation indices for unstimulated and Dex or EGF-stimulated cells. $n = 4$, *$p < 0.05$, two-tailed Welch's $t$-test.

inhibition (Supplementary Fig. 8e, f). Taken together, these results demonstrate that CK2 activation, likely through the phosphorylation of clock proteins such as BMAL1 at Ser90, plays a key role in initiating the synchronization of the multicellular circadian clock.

### Simulation analysis of the relationship between immediate BMAL1-Ser90 phosphorylation and synchronization

Our findings indicated that clock-synchronizing factors trigger the phosphorylation of BMAL1-Ser90 by CK2, which is then followed by the nuclear translocation of BMAL1. This leads to a subsequent surge in *Per2*, which mediates a phase shift and synchronization in the multicellular circadian clocks. To understand the significance of CK2-mediated BMAL1

phosphorylation in this process, we performed a numerical simulation to explore the relationship between phosphorylation of BMAL1, including Ser90 phosphorylation, and circadian clock oscillation.

The model we employed was based on the Kim-Forger model[21], a comprehensive model that facilitates the simulation of various molecular states and protein complexes involved in circadian oscillations (Supplementary Fig. 9a). In our simulation model, we hypothesized that BMAL1 Ser90 phosphorylation comprises a part of the BMAL1 phosphorylation essential for nuclear translocation of the BMAL1:CLOCK heterodimer to induce the transcriptional activation of E-box-containing genes, including *Per2* (Fig. 5a). This assumption was grounded in our previous findings that the level of BMAL1-Ser90 phosphorylation exhibits circadian fluctuation

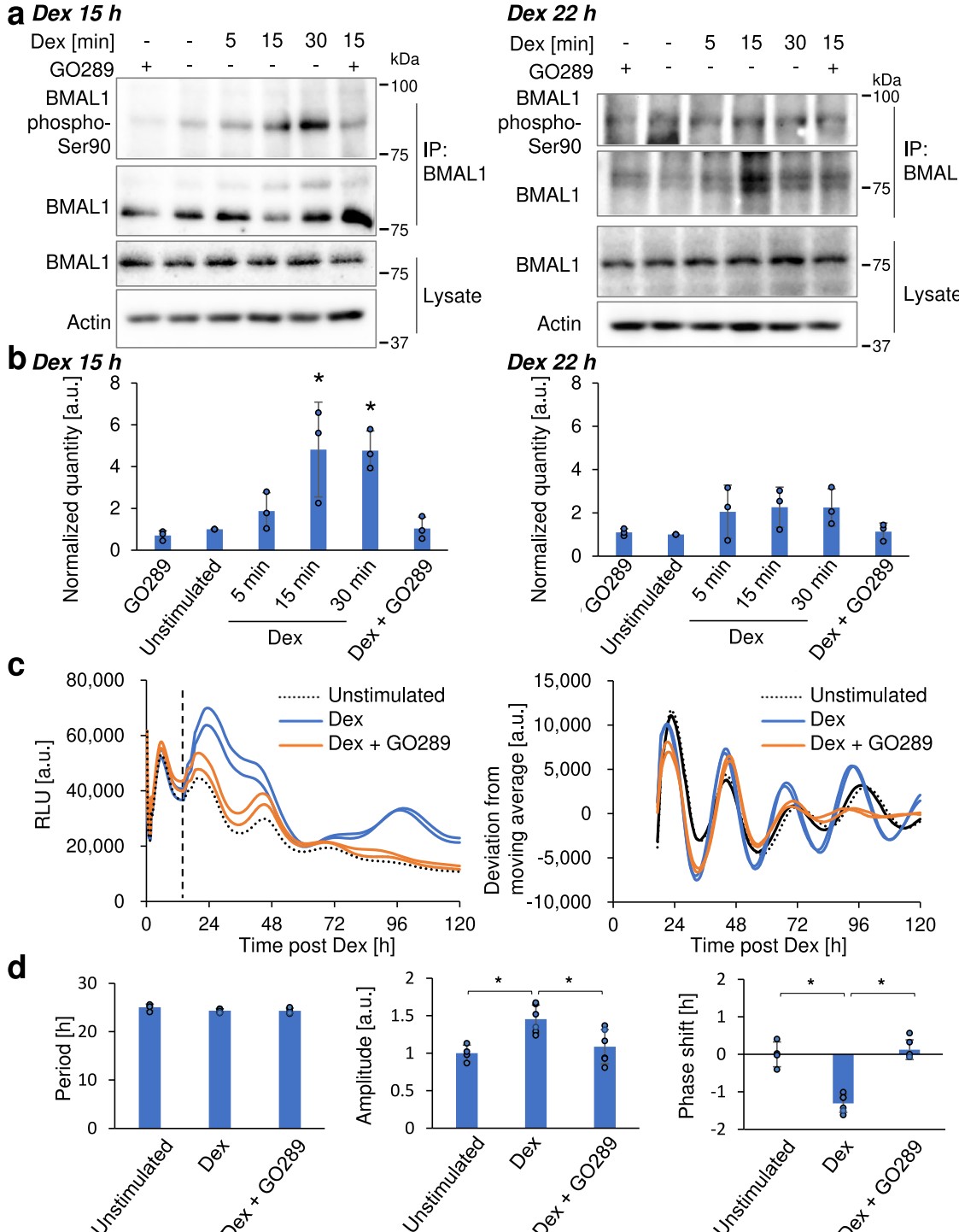

**Fig. 4 | CK2-mediated BMAL1-S90 phosphorylation immediately after clock synchronizing stimulation contributes to clock phase shift. a** Representative immunoblot images of BMAL1-Ser90 phosphorylation bands. Cells were stimulated with Dex in the presence or absence of CK2 inhibitor, GO289, and sampled at the indicated time post 15 h (Dex 15 h) or 22 h (Dex 22 h) after clock synchronization with Dex for immunoprecipitation using an anti-BMAL1 antibody. **b** Quantification of BMAL1-Ser90 phosphorylation levels. Phosphorylation levels were normalized to the total BMAL1 protein levels in each immunoprecipitated sample and further normalized to the unstimulated control. Actin was used as a loading control. Data represents the mean of three independent experiments. Error bar: SD, *$p < 0.05$, two-tailed *t*-test. **c** MEF Per2-Luc cells were pre-synchronized with Dex and re-

stimulated with Dex post 15 h following the initial Dex stimulation in the presence ("Dex + GO289") or absence ("Dex") of CK2 inhibitor, GO289. Cells were transiently stimulated for 60 min and returned to the original medium after stimulation. "Unstimulated" denotes a sample left unstimulated. The temporal luminescence profile of Per2-Luc was monitored. Representative Per2-Luc profiles, and normalized Per2-Luc profiles calculated by subtracting the moving average values, are shown. The dotted line indicates the time point when cells were re-stimulated. **d** Period and amplitude of the Per2-Luc luminescence profile following the second Dex stimulation. There are at least 4 replicates for each condition. ±: SD. *$p < 0.05$, Tukey HSD test.

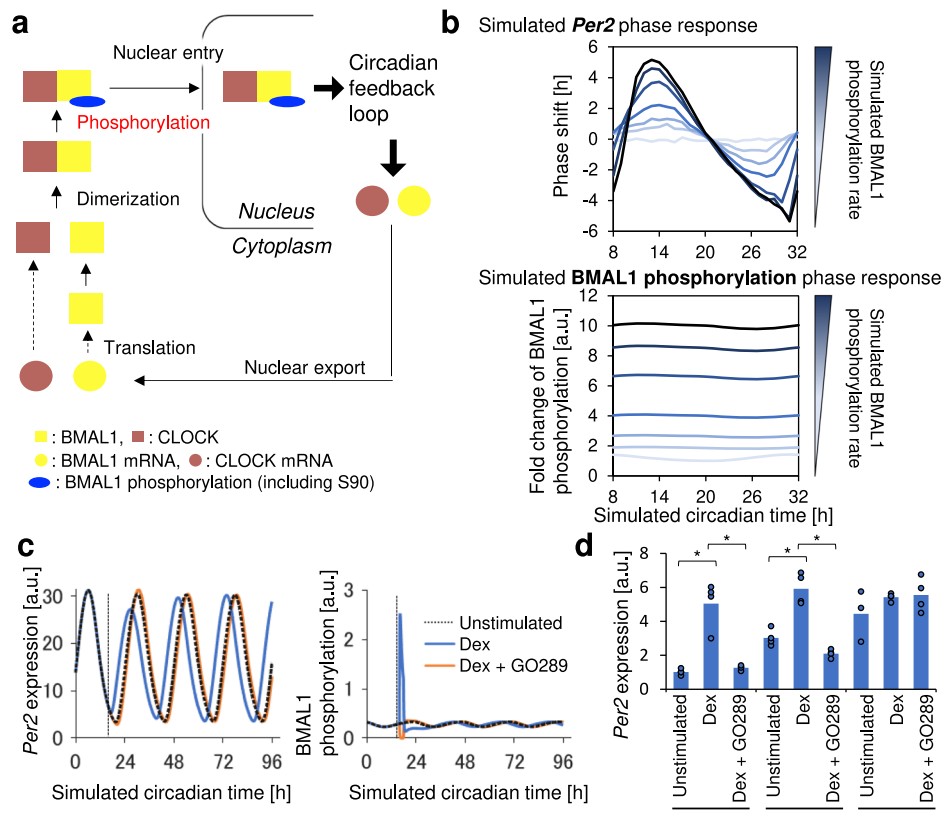

**Fig. 5 | Simulation of the effect of BMAL1-Ser90 phosphorylation on *Per2* phase shift. a** Schematic diagram showing the modified model used for the simulation based on the original model (Supplementary Fig. 9a). The model incorporates a mechanism where the phosphorylation rate constant of BMAL1 is modulated during synchronization. In the schematic diagram, only the scheme directly related to BMAL1 nuclear translocation is shown. Other clock components are included in the actual model. **b** Simulated circadian time-dependency of *Per2* phase shift and BMAL1 phosphorylation level change. Changes before and after a short-term increase in the BMAL1 phosphorylation level with different induction levels were simulated. The phase response curve of the simulated *Per2* expression was calculated by taking the mean *Per2* phase shift between simulations with and without BMAL1 phosphorylation induction. The fold change of BMAL1 phosphorylation at each circadian time was calculated as the ratio of the maximum phosphorylation level of phosphorylated BMAL1:CLOCK to the baseline phosphorylation level. **c** Simulation of Per2-Luc in response to CK2 inhibition. Dex stimulation was modeled as an increase in BMAL1 phosphorylation rate for a duration of 2 h. CK2 inhibition was modeled as a decrease in the rate of BMAL1 phosphorylation for 2 h. The dotted line indicates a time point when Dex ("Dex") or a combination of Dex and GO289 ("Dex + GO289") were applied. "Unstimulated" represents the simulation without Dex or GO289 treatment. **d** mRNA levels of the *Per2* gene following Dex stimulation. NIH-3T3 cells were first synchronized with Dex and re-stimulated with Dex in the presence or absence of CK2 inhibitor, GO289, at indicated time points. mRNA was collected 4 h after the second stimulation and quantified by qPCR. $n = 3$, error bar; SD, $*p < 0.05$, Tukey HSD test, not significant ($p > 0.05$) unless mentioned.

and that the non-phosphorylatable mutant of BMAL1-Ser90 fails to rescue the circadian *Per2* oscillation in BMAL1-KO MEF cells[14,15]. We found that our refined model was able to reproduce the circadian oscillation of core clock genes, including *Per1/2*, *Cry1/2*, *Bmal1*, and *Rev-Erb* (Supplementary Fig. 9b), validating its utility for investigating the relationship between BMAL1 phosphorylation and clock synchronization.

Using this model, we conducted a series of in silico experiments to assess whether BMAL1 phosphorylation serves as a universal cue for inducing a phase shift of circadian clock oscillation. We simulated the effects of a synchronizing factor as a transient increase in the phosphorylation rate of BMAL1. In this simulation, the model had two arbitrarily defined values: the magnitude of the phosphorylation rate constant change and duration (Supplementary Fig. 10a). Hence, we first simulated the impact of the level of rate change and duration on the maximum level of Ser90 phosphorylation induction (Supplementary Figs. 10b and 11). We found that for a given rate change, the maximum Ser90 phosphorylation level remained consistent regardless of the duration, except for a duration of 0.1 h. Considering the experimentally obtained BMAL1 Ser90 phosphorylation induction fold change (Fig. 4b), we set the magnitude to a 15-fold increase. We also confirmed that a simple, constant rate increase produced nearly identical outcomes to a complex pulse-like pattern, so we adopted the constant rate for simplicity (Supplementary Fig. 10c). We note that although longer duration

and higher levels of phosphorylation rate change tended to induce a larger induction level of *Per2*, as well as a phase shift of *Per2* oscillation (Supplementary Fig. 11), duration had a limited influence on the predetermined magnitude of a 15-fold increase in the phosphorylation rate. Consequently, the duration was set to 2 h, which corresponds to the duration of the synchronization experiment.

The simulation revealed that a short-term increase in the rate of BMAL1 phosphorylation induced a phase shift in *Per2* oscillation, similar to that caused by the direct upregulation of E-box genes including *Per2* (Fig. 5b and Supplementary Fig. 12a, b). These results suggest that a transient increase in the BMAL1 phosphorylation level could induce a phase shift of circadian clock. Then, we sought to identify the circadian time-dependent relationship between BMAL1 phosphorylation level and a phase shift in *Per2* oscillation. For this purpose, we simulated a shift in *Per2* expression induced by arbitrary rates of BMAL1 phosphorylation at various circadian times. Our simulations revealed a pronounced circadian time-dependency in the magnitude of the *Per2* phase shift (Fig. 5b and Supplementary Fig. 11a). Additionally, we found that the simulated foldchange in BMAL1 phosphorylation level had a slight circadian time dependency, with a slight peak at around circadian time 13 h (Fig. 5b). This simulation result aligns with our findings that the change in BMAL1-Ser90 phosphorylation level was more pronounced at circadian times when *Per2* expression was relatively low (Fig. 4a).

In addition, we used the *Per2* expression in our simulation as an indicator of Per2-Luc to examine whether the simulation model can recapture the behavior of Per2-Luc following Dex stimulation, both in the presence or in the absence of CK2 inhibitor, GO289 (Fig. 5c). We assumed that the experimentally observed Per2-Luc luminescence signal is directly proportional to the simulated expression of *Per2* and that Dex stimulation induces a short-term increase in the BMAL1 phosphorylation rate, which reflects the experimentally observed BMAL1-Ser90 phosphorylation level change (Fig. 4b). The simulation result revealed that the Dex-induced phase shift was suppressed under the condition where BMAL1-Ser90 phosphorylation was inhibited by GO289 treatment, which was modeled by setting the rate constant of BMAL1 phosphorylation to zero during synchronization. Moreover, when we assumed that simulated *Bmal1* was proportional to Bmal1-Luc reporter signal, the model successfully reproduced the observed suppression of the Dex-induced phase shift of Bmal1-Luc upon inhibition of BMAL1 Ser90 phosphorylation (Supplementary Fig. 12c).

To further validate the relationship between BMAL1 phosphorylation and *Per2* surge, we experimentally examined the circadian time-dependent effects of CK2 inhibition on *Per2* expression. As a result, we found that the suppression of *Per2* surge by inhibition of CK2 was most pronounced at Dex 15 h and 18 h, but less pronounced at Dex 22 h (Fig. 5d).

Taken together, these findings suggest that phosphorylation of BMAL1 including Ser90 and subsequent nuclear BMAL1 accumulation during the initial stages of the clock synchronization play critical roles in inducing *Per2* expression and driving the phase shifts of circadian clocks in fibroblasts.

## Discussion

Nuclear entry of BMAL1 and BMAL1-dependent CLOCK, forming the BMAL1:CLOCK heterodimer, is crucial for the transactivation of *Per1/2* clock genes. An acute surge in *Per1/2* expression is recognized as a hallmark event following clock synchronization[10,11]. Therefore, we hypothesized that the immediate nuclear accumulation of BMAL1 and CLOCK comprise a pivotal switch for clock synchronization. This study provides a detailed analysis of the mechanism underlying immediate BMAL1 nuclear accumulation during clock synchronization is response to various external stimuli including glucocorticoids, mitogens, 2nd messengers, $Ca^{2+}$ ionophores, and heat shock stress, in mouse fibroblast cells.

We initially investigated BMAL1 nuclear accumulation in response to various synchronizing factors in desynchronized cells. We found that diverse clock synchronizers consistently induced immediate nuclear accumulation of BMAL1 and CLOCK. Previous studies have shown that BMAL1 rapidly enter nucleus upon serum stimulation within 30 min post stimulation, and that a nuclear localization signal-deficient BMAL1 mutant blunts rhythmic expression of Per2-Luc[9], supporting the notion that nuclear accumulation is critical in clock synchronization. While a previous study reported a lack of BMAL1:CLOCK nuclear accumulation in response to EGF stimulation[9], our observations, using a broader spectrum of synchronizing factors and assessing BMAL1 at endogenous expression levels, suggest that BMAL1 nuclear accumulation may be a more general phenomenon during clock synchronization. Indeed, observation of nuclear BMAL1 accumulation suggested commonality of the BMAL1-ISR phenomenon across various biological contexts, including cell types and synchronization stimulation. However, since our focus was only on cell types and stimulation exhibiting circadian clock synchronization and oscillation, we could not comprehensively verify this phenomenon to validate its universality. Nevertheless, ISR could be one of the important molecular mechanisms governing synchronization, with an expectation of a new molecular target for inducing circadian clock synchronization. Further studies are warranted to determine whether nuclear accumulation of BMAL1 or enhanced transcriptional activation capacity by nuclear BMAL1 during early clock synchrony is a universal feature of clock synchrony in different tissues and in vivo.

Based on our findings, we propose that nuclear accumulation of BMAL1, and CLOCK is a key event in circadian synchronization, at least in cultured fibroblasts. The subcellular localization pattern of BMAL1 is regulated by its phosphorylation states and its dimerization with CLOCK or NPAS2[22]. For the roles of BMAL1 in the cytoplasm, phosphorylation of BMAL1 at Ser42 by AKT or S6K, which share the same phosphorylation site motif (RXXS*/T*), has been implicated to promote BMAL1 cytoplasmic retention where BMAL1 play roles for translation and metabolism[23,24]. Notably, it has been reported that BMAL1 phosphorylation is promoted for CLOCK-dimerized form of BMAL1[16,25], suggesting a potential link between BMAL1 phosphorylation and its transactivating role with CLOCK in the nucleus. Additionally, there are other mechanisms that control the localization of BMAL1, such as BMAL1 interaction with mRNA export factor RAE1[26]. Since BMAL1-Ser90 phosphorylation drives circadian oscillations of clock gene expression by promoting BMAL1 accumulation in the nucleus[14], it is plausible that transient BMAL1 phosphorylation by CK2 including phosphorylation at Ser90 immediately following clock synchronizer stimulation influences the phase of the circadian rhythm by modulating BMAL1 nuclear accumulation and transcriptional activation of clock genes.

Phosphorylation of BMAL1 at Ser90 affects some of the key events in the regulation of *Per2* expression driven by BMAL1:CLOCK. In contrast to BMAL1 WT, the phospho-deficient mutant of BMAL1-Ser90 (BMAL1-S90A) exhibits weak nuclear accumulation and fails to rescue the rhythmic *Per2* expression in BMAL1-KO MEFs, suggesting that Ser90-phosphorylation is essential for BMAL1:CLOCK transactivation in the nucleus[14]. Furthermore, we reported that GO289, a highly selective inhibitor of CK2 responsible for BMAL1-Ser90 phosphorylation, prolonged circadian period, in a dose dependent manner and almost completely dampening oscillations at high doses[20]. In line, we found a circadian-time dependent *Per2* surge that was suppressed by repressing BMAL1 nuclear accumulation by inhibiting BMAL1-Ser90 phosphorylation by CK2. CK2 is also suggested to modulates circadian clock by regulating the degradation of PER2[27,28]. Therefore, the result of GO289 inhibitor assays may be a consequence of combined result of CK2-mediated phosphorylation of BMAL1 and PER2. Collectively, these observations strongly suggest that immediate BMAL1 phosphorylation, and subsequent BMAL1:CLOCK nuclear accumulation is one of the switching events for clock synchronization.

We used both immunocytochemical methods and live cell imaging with a monomeric fluorescent protein mutant mVenus fused to BMAL1 to observe changes in the nuclear and cytoplasmic distribution of BMAL1. While both methods resulted in the detection of immediate nuclear accumulation of BMAL1 as early as 15 min post clock-synchronizing stimuli, we detected slightly different localization patterns of BMAL1, with live cell imaging detecting less cytoplasmic localization of BMAL1. Although we used a *Bmal1* promoter-driven probe for the live imaging, there may be a potential difference in expression level that causes the localization difference. In addition, the possibility of artifacts from the fluorescent protein mVenus attached to BMAL1 with the linker should also be considered, as revealed in Supplementary Fig. 6c. Nevertheless, both immunostaining and live cell imaging observed nuclear accumulation of BMAL1 upon clock synchronizer stimulation, suggesting that these results are not necessarily inconsistent with the BMAL1-ISR concept.

In this research, we employed Dex treatment, a potent glucocorticoid with an established role as a universal clock synchronizing agent in vivo, as a model system to synchronize the multicellular circadian clock. We applied a second Dex stimulation at 15 h, 18 h and 22 h after the initial Dex stimulation to study the time-dependence of BMAL1-ISR. These time points were chosen based on the observation that the nucleocytoplasmic distribution of BMAL1 is predominantly cytoplasmic around 16 h, and primarily nuclear around 22 h post serum shock[7]. Indeed, we observed similar subcellular distribution of BMAL1 without the second Dex stimulation. To further characterize the BMAL1-ISR, we focused on Dex 15 h condition for CK2 inhibition and simulation experiments. Compared to Dex 18 h, Dex 15 h is a time point when *Per2* expression is at its lowest, and BMAL1-Ser90 phosphorylation is in the ascending phase[15]. Furthermore, the phase shift observed in Bmal1-Luc was more pronounced at Dex 15 h compared to the phase shift at Dex 18 h[29], suggesting that Dex 15 h is the optimal circadian

time for observing the effect of BMAL1 Ser90 phosphorylation on circadian clock synchronization.

In conclusion, this study provides evidence that the phosphorylation of BMAL1 by CK2, which promotes nuclear accumulation and likely enhances the transcriptional activity of BMAL1:CLOCK, occurs immediately after various clock-entrainment stimulation. This BMAL1-ISR phenomenon is involved in the acute *Per2* surge and subsequent clock phase shift, acting as a switching signal linking internal clock oscillation and synchronization (Supplementary Fig. 13).

## Materials and methods
### Cell culture
Mouse fibroblast NIH-3T3 cell (RIKEN cell bank, Japan) clones harboring Bmal1-luciferase (Luc) or Per2-Luc were established in the previous study[15]. C6 cell (RIKEN Cell Bank, Japan), mouse fibroblast MEF cell (obtained from the previous study[30]), mouse myoblast C2C12 cell (CRL-1772, ATCC), and plat-E cells (Cell Biolabs Inc.) were cultivated with Dulbecco's modified eagle medium (D-MEM, Nacalai Tesque) supplemented with fetal bovine serum (FBS, Gibco) and 1% penicillin/streptomycin (Gibco). NIH-3T3 cell ablated of functional BMAL1 (BMAL1-KO) were generated using the CRISPR-Cas9 system as previously described[31]. Clock-synchronizing treatments were performed using near confluent cell cultures with the concentration and duration of stimulation as described in the figure legends. Transfection of a probe DNA plasmid to plat-E cells was performed using the TransIT-LT1 transfection reagent (Mirus bio) following the manufacturer's protocol. For the generation of a stable cell line expressing the BMAL1 translocation reporter, retrovirus infection was conducted using a retrovirus produced in plat-E cells using polybrene (hexadimethrine bromide, Sigma), at a final concentration of 0.24 mg mL$^{-1}$.

### Confocal imaging analysis for immunofluorescence of clock proteins
NIH-3T3 WT (without Bmal1-Luc or Per2-Luc) and BMAL1-KO (with Bmal1-Luc) cells were treated with each clock synchronizer or left unstimulated (negative control). Cells were then fixed with 4% paraformaldehyde and immunostained with anti-BMAL1 (1:250, rabbit polyclonal, clone Nt[7]) and anti-CLOCK (1:500, mouse monoclonal, MBP #D333-3 clone CLSP3[25]), or rabbit polyclonal (1:1000, Cell Signaling Technology #5157) antibodies. AlexaFluor488 conjugated anti-rabbit IgG (1:250, Thermo fisher #A-11008), AlexaFluor647 conjugated anti-rabbit IgG (1:1000, Thermo fisher #A-21245), or AlexaFlour568 conjugated anti-Mouse IgG (1:1000, Thermo fisher #A-11004) antibodies were used as a secondary antibody. DAPI (4',6-diamidino-2-phenylindole, Thermo fisher #D-1306) or Hoechst33342 (Thermo Fisher #H-3570) was used for nuclear staining. Immunostained BMAL1 and CLOCK localization were visualized by confocal imaging using a Carl Zeiss LSM-510 META microscope. Images were quantified using a custom Fiji-plugin macro, described in Supplementary Fig. 1. Briefly, the nuclei for each cell were segmented using the nuclear-stained image using a binarization program. For the cytoplasmic region, the boundary of the segmented nucleus image is expanded by 3 pixels by dilation. These segmented images were used as a mask image to obtain fluorescence intensity for each region by image calculation. Nucleus-to-cytoplasmic ratios were calculated by dividing the average nuclear fluorescence intensity by the average cytoplasmic intensity.

### Plasmid construction
To avoid dimerization artifacts, Venus fluorescent protein was converted to monomeric mVenus by a genetic point mutation replacing Ala207 (GCC) with Lys (AAA), using KOD-Plus-Mutagenesis Kit (TOYOBO). A mouse *Bmal* promoter-driven BMAL1 fused with mVenus (mVenus-BMAL1) reporter was generated by ligating mVenus to the N-terminal of BMAL1, followed by insertion of linkerA (GSAGSAAGSG EFGSAGSAAG SGEFEF) or linkerB (GGSGGSGSAG SAAGSGEFEF), respectively using PCR amplification and enzymatic digestion followed by ligation of the DNA

fragments. H2B-mKate2 was generated by ligating H2B to the N-terminal of mKate2 fluorescent protein. These reporters were subcloned into the pMX vector for retrovirus transduction. Plasmids newly generated in this study, along with their maps and nucleotide sequences, will be available at the Addgene repository (#249336, #249337, #249338) when the paper is published.

### Real-time bioluminescence monitoring and data processing
Cells were synchronized with dexamethasone treatment (10 nM, 2 h). Real-time bioluminescence was monitored using Kronos Dio (Version 2.10.230, ATTO, Japan) with acquisition intervals of 10 min. For bioluminescence monitoring, the culture medium was supplemented with 0.1 mM D-luciferin (Wako, Japan) and 10 mM HEPES. For pulsed CK2 inhibition experiments, cells were transiently stimulated with 10 μM GO289 (kindly donated by Dr. Hirota)[20] for 60 min. In this experiment, culture medium (3 mL) was removed, of which 1 mL was mixed with Dex (100 nM) or Dex (100 nM) with GO289 and placed back into each dish. After 60 min of incubation, the stimulant-containing medium was replaced with the remaining medium that are kept at 37 °C. The number of replicates is indicated in each experiment. "Deviation from the moving average" in the *Y* axis represents the raw values that were detrended by subtracting a moving average of 24-h window according to the program within the detector (Kronos; ATTO, Japan). Circadian rhythmicity were analyzed using the R packages "lomb"[32] and "cosinor2"[33] to determine the period, amplitude, and phases.

### Live-cell imaging of BMAL1 localization change
Live-cell imaging of BMAL1 localization were performed using an epifluorescence microscopy system equipped with a luminescence microscope (IXplore Live for Luminescence; Olympus). mVenus-BMAL1 and H2B-mKate2 fluorescence was visualized using U-FYFP and U-FMCHE filter cubes (Olympus), respectively. Images were acquired with a 20× air/dry objective (NA 0.95; Olympus Corp.) and an EM-CCD camera (iXon Ultra 888; Andor). Cells were synchronized with EGF (100 ng mL$^{-1}$) or Dex (100 nM) stimulation and observed under a microscope at 3 min intervals with acquisition time of 2 s. Culture medium used for imaging was D-MEM supplemented with 10 mM HEPES.

### Single cell tracking analysis of BMAL1 localization change
Single-cell tracking of a nucleus-to-cytoplasm fluorescent intensity ratio was calculated by the following procedure. First, cellular segmentation for the nucleus and cytoplasm was performed using Cellpose[34]. To segment the image, BMAL1 reporter images were used as an image for the entire cell, and H2B-mKate2 images were used as an image for the nucleus. By using a segmentation mask generated by the Cellpose, BMAL1 reporter images were separated into a cytoplasm image and a nucleus image. Single-cell tracking was performed using the TrackMate plugin in Fiji software[35], according to the developer's protocol. For the tracking, H2B-mKate2 image was used. Using the coordinates of tracked single-cell trajectories, intensities of the nucleus and cytoplasm were quantified using the "wand tool" in Fiji software[36]. Relative nucleus intensity was calculated as a ratio of nucleus fluorescence intensity to whole cell fluorescence intensity. The positive rate of cells for BMAL1 nuclear localization was determined by first calculating the maximum fold change in the nuclear-to-cytoplasmic ratio for each cell. Subsequently, the threshold was defined as the 75th percentile of the ratio changes for the unstimulated control. Cells exhibiting a maximum fold change in nuclear-to-cytoplasmic ratio above this threshold were designated as positive.

### Local correlation analysis
To calculate correlation among cells, we employed Gaussian graphical models, a graphical lasso, which is implemented with the R package "glasso"[37]. The regularization parameter ($\rho$) in the graphical lasso, which controls the sparsity of the correlation network, is arbitrary set to be 0.013. The graphical lasso identifies temporally correlated nuclear BMAL1

translocation among the analyzed cells. The highly correlated cells were identified as an "edge" that connects the two cells. To determine the neighboring cells, the Delaunay triangulation algorithm implemented with the R package "RTriangle"[38] was used. The neighboring cells were defined as cells with an edge that connects the two cells by triangulation. For this triangulation, an average of cellular coordinates within the tracked trajectories were used as a coordinate of each cell. The local correlation index was defined as the ratio of the number of correlated cells to the number of neighboring cells. Therefore, the local correlation index was calculated as the summation of edges in the correlated cells within the neighboring cells, normalized by the summation of edges in the neighboring cells. The global correlation index is defined as the ratio of the number of correlated cells to the number of all the analyzed cells. Therefore, the global correlation index was calculated as the total number of edges identified in correlated cells normalized by the number of all the possible edges among observed cells. To calculate the time evolution of these indices, the time course data within the time window from 0 min to the time of each point was used.

### Immunoblot and immunoprecipitation assays

Cells cultivated in a culture dish were exposed to Dex (100 nM) or EGF (100 ng mL$^{-1}$) and returned to the incubator until the predetermined time point for sample collection. For CK2 inhibition samples, cells were treated with GO289 (10 μM). At the sampling time point, cells were washed twice with ice-cold PBS and lysed with an NP-40 lysis buffer {10 mM Tris-HCl (pH = 7.4), 150 mM NaCl, 5 mM EDTA, 50 mM NaF and 0.5% NP-40} supplemented with a protease inhibitor cocktail (Complete, Roche) and a phosphatase inhibitor cocktail (PhosSTOP, Roche). The lysed samples were then centrifuged at 4 °C at 15,000 rpm for 20 min. For immunoprecipitation assays, the supernatant was collected, and an NP-40 lysis buffer was added to dilute the sample to 1 mL. The diluted sample was then incubated with 0.5 μg anti-BMAL1-Nt antibody[7] for 4 h. Then 30 μL of G-Sepharose beads (GE Healthcare) was added and further incubated for 1 h. The protein Sepharose beads were collected and washed with PBS buffer three times and diluted with lysis buffer. The loading samples for SDS polyacrylamide gel electrophoresis (SDS-PAGE) were prepared by the addition of 0.2 equivalent of 5x sampling buffer {250 mM Tris-HCl (pH 7.6), 10% SDS, 25% Glycerol, 5% 2-mercaptoethanol, 0.02% bromophenol blue} to the supernatant of the centrifuged mixture. Immunoblotting was performed at 4 °C overnight using specific primary antibodies against the target protein including primary antibodies against BMAL1 (1:1000), phospho-Ser90 BMAL1 (1:500), Histones (1:1000, Chemicon, # MAB052), H3 (1:2000, abcam, #ab1791) and actin (1:5000, Sigma AC-15 clone). Anti-BMAL1 and phospho-Ser90 BMAL1 antibodies were previously generated in our studies[7,14]. Anti-rabbit IgG (1:5000) and anti-mouse IgG (1:5000) antibodies labeled with horse radish peroxidase (GE Healthcare) were used as secondary antibodies. Chemiluminescence from the immunostained bands was detected with the LuminoGraph III Lite detection system (Ver. 1.3.8, ATTO), and the band intensities were quantified using Fiji software (ImageJ version 1.54p).

### Nuclear fractionation

Nuclear and cytosolic fractions at the indicated time points were prepared as previously described[39]. Briefly, at the sampling time point, cells were washed twice with ice-cold PBS and lysed with a ChI buffer (40 mM HEPES (pH 7.5), 0.1% NP-10, 1 mM dithiothreitol, 0.32 M sucrose, 10 mM KCl, 3 mM EDTA) supplemented with a protease inhibitor cocktail (Complete, Roche) and a phosphatase inhibitor cocktail (PhosSTOP, Roche). Cells were homogenized using a 27-gauge needle and incubated on ice for 20 min. The lysed samples were then centrifuged at 4 °C at 4000 rpm for 5 min, and washed with ChI buffer three times. The supernatant was collected as a cytoplasmic fraction. The precipitate was lysed in ChII buffer (40 mM HEPES (pH 7.5), 1% NP-40, 1 mM dithiothreitol, 0.4 M NaCl, 3 mM EDTA) supplemented with a protease inhibitor cocktail (Complete, Roche) and a phosphatase inhibitor cocktail (PhosSTOP, Roche) to obtain a nuclear fraction.

### Quantitative PCR

For quantification of the circadian *Per2* gene, mRNA was collected from NIH-3T3 cells exposed to Dex using Trizol reagent (Invitrogen). Collected mRNAs were reverse transcribed using PrimeScript Reverse Transcriptase (TakaraBio, Japan). The cDNA fragments were amplified using THUNDERBIRD SYBR qPCR Mix (TOYOBO, Japan) following the manufacturer's protocol. The fluorescence amplification curve was detected by a Thermal Cycler Dice Real Time System II (Ver. 5.11B, TakaraBio, Japan). To quantify gene expression, values were first normalized to *β-actin* expression, and then to the abundance at time 0. The following primers were used for the analysis: *Per2*, 5'- CAG GTT CCG CCC CGC CAG TAT -3', and 5' - GTC GCC CTC CGC TGT CAC ATA G -3'; *β-actin*, 5'-ACT GCT CTG GCT CCT AGC AC -3' and 5'-ACA TCT GCT GGA AGG TGG AC-3'.

### Mathematical modeling

The model used in this study is based on the detailed model originally proposed by Kim and Forger[21]. The original model comprised 181 variables, which included monomer proteins and complexes of PER1, PER2, CRY1, CRY2, BMAL1, CLOCK/NPAS2, REV-ERBs, CK1, and GSK3β. The interactions among these variables were characterized using systems of ordinary differential equations (ODEs) and explicit mass kinetics. To incorporate the observed effects of BMAL1 Ser90 phosphorylation, we introduced an additional coefficient to modulate the rate constant of BMAL1 phosphorylation within the model. Since BMAL1 phosphorylation occurs post-heterodimerization with CLOCK in the original model, we considered phosphorylated BMAL1:CLOCK as the molecular species representing the experimentally observed changes in BMAL1 Ser90 phosphorylation levels.

For the simulation of the effect of BMAL1-Ser90 phosphorylation on *Per2* expression, we first evaluated the optimal coefficient for the rate constant for BMAL1 phosphorylation (Phos rate) by a factor ranging from 1 to 50, and also the duration of Phos rate increase ranging from 0.1 h to 8 h. The range for Phos rate was determined based on the report that activation change of the potential responsible kinase for BMAL1 Ser90, CK2, would occur in the order of 10[40], and that positive effect would be on the Phos rate. The range for duration was based on the assumption that an increase in Phos rate would occur transiently, but could range from short to long duration. We systematically applied a combination of Phos rate increase (1, 2, 3, 5, 10, 15, 20, 30, and 50) and duration (0.1, 0.2, 0.5, 1, 2, 4, 6, and 8 h) at every 1 h between simulated circadian time of 8–32 h as a simulated synchronization effect. Fold change of BMAL1 phosphorylation and *Per2* expression was defined as the ratio of maximum induction levels after simulated synchronization, and *Per2* phase shift was calculated as the phase shift of *Per2* oscillation compared to simulation without any Phos rate change. To model the effects of CK2 inhibition during clock synchronization, we applied a factor of 15 to the phosphorylation rate of BMAL1 for a time span of 2 h to simulate Dex-induced synchronization. This was based on the maximum fold change in phosphorylation levels observed experimentally after Dex stimulation. To simulate the effect of CK2 inhibitor treatment, we transiently set the rate constant for the conversion of non-phosphorylated BMAL1 to phosphorylated BMAL1 to zero for a period of 2 h. All simulations were performed using MATLAB version R2023a (MathWorks).

### Statistics and reproducibility

All experiments were conducted with a specifically chosen sample size, and the sample number for each experiment is described in the figure legend. No statistical sample size calculations were performed. Each replicate experiment was performed by applying the same technical protocol to separately cultured cells with different passage numbers. Randomization was not relevant for data included in the manuscript. Error bars represent the standard deviation as described in the figure legends. For boxplots, a box is used to indicate the positions of the upper and lower quartiles; the interior of this box indicates the interquartile range, which is the area between the upper and lower quartiles. Whiskers are extended to the extrema of the

distribution to 1.5 times of the interquartile range. Statistical values ($p$ values) were obtained from statistical tests described in figure legends. For comparison of multiple conditions, either Tukey's honestly significant difference test or Dunnet test with Bonferroni adjustment was applied to assess the mean differences among the conditions. The statistical value below 0.05 ($p < 0.05$) is considered to be statistically significant. Exact $p$ values are provided as Supplementary Table 1. Statistical tests were performed using R version 4.4.0 and RStudio version 2023.06.1 + 524.

## Artificial intelligence (AI)
Grammarly (Grammarly Inc.) was partly used for English proofreading for grammatical errors. No original sentences were generated using AI chatbots.

## Reporting summary
Further information on research design is available in the Nature Portfolio Reporting Summary linked to this article.

## Data availability
The numerical source data for all the graphs and charts are available in Supplementary Data 1. Uncropped images for Fig. 1a and Supplementary Fig. 4a are provided in Supplementary Fig. 14. The uncropped blots related to Figs. 2d, 4a and Supplementary Fig. 8a, c are provided in Supplementary Figs. 15–17, respectively. All other data are available from the corresponding author on reasonable request.

## Code availability
The codes for the simulation and analysis are deposited at Zenodo [https://doi.org/10.5281/zenodo.17568955].

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

## Acknowledgements
We thank Drs. T. Hirota, Y. Shigeyoshi, and P. Sassone-Corsi for providing us with reagents. T.T. discloses support for the research of this work from JSPS KAKENHI [grant number 19K07288]. G.K. discloses support for the research of this work from JSPS KAKENHI [grant number JP22K14779].

## Author contributions
T.T. and G.K. designed research; T.T. and G.K. performed research; T.T. and G.K. analyzed data; H.Y., S.K., and Y.F. contributed materials; T.T. and G.K. wrote the original draft of the manuscript; T.T., G.K., H.Y., A.N., T.O., and K.T. reviewed and edited the manuscript. All the authors approved the final version of the manuscript.

## Competing interests
The authors declare no competing interests.
