## [Transparent Peer Review file · Communications Biology]

Immediate nuclear accumulation of BMAL1 to regulate cellular circadian clock synchronization

Corresponding Author: Dr Teruya Tamaru

Version 0:

Reviewer comments:

Reviewer #1

(Remarks to the Author)

This manuscript examines the role of BMAL1-Ser90 phosphorylation in nuclear translocation during circadian clock synchronization and introduces BMAL1 Immediate Synchronization Response (ISR) as a proposed universal mechanism across synchronizing stimuli. However, the study largely reiterates findings already established in the field without presenting significant novel insights. Furthermore, the data provided, particularly the live-cell imaging movies (I cannot see any described changes), fail to convincingly support the described phenomena. Besides, the study is confined to NIH-3T3 fibroblasts. While the manuscript claims BMAL1-ISR is a universal mechanism, it lacks validation in other cell types, tissues, or in vivo models to substantiate this claim.

Reviewer #2

(Remarks to the Author)

Tamaru and colleagues hypothesize that the nuclear accumulation of both BMAL1 and CLOCK after chemical synchronization in NIH 3T3 cells is a signal for molecular clock synchronization.

The authors premise their hypothesis on a previously published observation of BMAL1 rhythms in cellular distribution after serum starvation in NIH 3T3 cells such that BMAL1 was predominantly found in the nucleus 4 hours after synchronization, more so in the cytoplasm after 16 hours, and again nuclear after 24 hours. While others have shown that PER1/2 expression increases as a result of various entrainment signals, including serum shock, the authors here posit that nuclear accumulation of the positive elements of the transcriptional feedback loop is a necessary predicating step for the PER1/2 'surge'. This hypothesis is intriguing for several reasons. If supported, this would introduce more than one novel target for modulating synchronization, which lies at the core of clock resetting; not just at the molecular level, but potentially for the whole organism.

The authors tested their hypothesis using confocal imaging and subtractive imaging algorithms to compare the ratio of nuclear:cytoplasmic expression of BMAL1 and CLOCK after synchronization by various methods. While the approach is appropriate, the images provided are difficult to immediately interpret. I would recommend providing a merge of DAPI and BMAL1 / CLOCK signals to highlight the changes in nucleocytoplasmic ratios in the various preps. As it stands, the Western blots of separated nuclear and cytoplasmic fractions are more compelling and may be more appropriate as a main figure.

The manuscript also investigates whether the endogenous clock affects BMAL1-based ISR and shows that there is a 'time-of-day' difference in the response. This does make sense, as the phase of molecular rhythms would be established by the initial synchronization signal, and it would be expected that nuclear BMAL1 would be lowest at around 12hrs post synchronization and highest after ~24 hours. I'm not sure this warrants a main figure to be honest and perhaps should trade places with the Western blots mentioned above.

BMAL1 nuclear accumulation was further confirmed in single cells via timelapse imaging of Bmal1::luc reporter in BMAL1-deficient cells. The authors highlight cells that undergo 'ISR response', but this does not seem universal across all cells, which begs the question of how important this is if the ensemble signal is rhythmic (and therefore synchronized). Can the authors quantify how many cells were positive for ISR, and what does this mean in terms of global synchronization?

The authors hypothesize that phosphorylation of Ser90 by CK2 – a prior discovery by the authors – is crucial to BMAL1 nuclear entry in ISR. They show immediate increase in Ser90 phosphorylation after synchronization, which is inhibited by a CK2-specific inhibitor. Importantly, they show that CK2 phosphorylation is needed for the Dex-induced phase shifts in Per2 and Bmal1 that allow for synchronization.

Finally, the authors conducted in silico experiments based on the Kim-Forger computational model. While the results of these experiments all seem to align with the cell-models, the utility of the model is uncertain, mainly due to several parameters of the model being 'arbitrary' or, as stated by the authors, adjusting the BMAL1 phosphorylation rate constant coefficient by a factor of 1 – 50. There may be interest in how this value affects the simulation; is the rate of BMAL1 phosphorylation always constant? The authors should provide more comprehensive data, including the values that did not align with their results, and provide an explanation in terms of the mechanism they propose for why their specified values 'worked'.

Overall, the manuscript is presented well, there are no glaring grammatical issues, and the methods are sensibly described. The simulation methods should be more clearly described when referring to the input values. The discussion of the results is in-line with the results and notes the limitations of the study.

Reviewer #3

(Remarks to the Author)

This paper investigates how circadian clocks synchronize at the cellular level. The authors focus on the localization patterns of clock proteins BMAL1 and CLOCK in NIH-3T3 fibroblasts. They show that synchronized nuclear accumulation of BMAL1 and CLOCK occurs immediately after clock-resetting treatments (ISR). The study further shows that phosphorylation of BMAL1 at Ser90 by CK2, which promotes its nuclear accumulation, is also rapidly increased during this response. Inhibiting CK2 partially suppressed the acute increase of Per2 and disrupted clock resetting. Computational simulations supported the idea that increased BMAL1 phosphorylation and nuclear localization can reset the circadian clock.

1. The data presented in Figure 1 is not very convincing. The representative immunohistochemical images don't show much difference in signal between the different conditions. However, it seems that the unstimulated WT condition has significantly more cells compared to the other conditions, which may indicate that Dex caused cell death. If this is not the case, please choose better images that more accurately reflect the analyzed data.
2. Figure 1B – It seems that the Clock Nuc/Cyt ratio changes in response to longer treatment with Dex, while BMAL1 does not exhibit a similar time-dependent effect. Please explain.
3. Similarly to comment 1, there seems to be no change in signal between stimulated and unstimulated cells, which does not align with the analysis on the right side of Figure 2a. Please address this accordingly.
4. Supplementary Figure 3 – There is no evident change in signal between the conditions in the representative images, which would indicate immediate nuclear BMAL1 accumulation compared to control.
5. Figure 4 – For better accuracy, BMAL1 should be isolated from the nuclear fraction rather than cellular BMAL1. Hence, the housekeeping gene should be a nuclear marker and not actin.

Version 1:

Reviewer comments:

Reviewer #1

(Remarks to the Author)

All my concerns have been addressed.

Reviewer #3

(Remarks to the Author)

Thank you for addressing my comments, I am satisfied with the revised version of the manuscript.

RESPONSE TO THE REVIEWERS

We are grateful to all three reviewers for their constructive comments and specific suggestions on our manuscript. We highly appreciate the reviewer's suggestions, which significantly improved our manuscript. Below are our point-by-point responses to the reviewer's comments. To clarify the discussion, the reviewers' comments are shown in blue, and our responses are in black. The revised parts of the manuscript and the SI are highlighted in yellow.

Reviewer #1 (Remarks to the Author):

This manuscript examines the role of BMAL1-Ser90 phosphorylation in nuclear translocation during circadian clock synchronization and introduces BMAL1 Immediate Synchronization Response (ISR) as a proposed universal mechanism across synchronizing stimuli. However, the study largely reiterates findings already established in the field without presenting significant novel insights.

Answer: We appreciate the reviewer's concerns about our experimental design. Our study is consistent with the preceding research examining the functional relevance of BMAL1 Ser90 phosphorylation in clock organization. Although our group and others have previously demonstrated that nuclear translocation of BMAL1 is crucial for synchronization and that Ser90 phosphorylation of BMAL1 is essential for maintaining circadian clock oscillation, there has been a lack of evidence concerning whether Ser90 phosphorylation is also essential for the clock reset, probably by controlling BMAL1 translocation as well as the molecular events that are triggered by the synchronization factors. The objective of this study was to comprehend the mechanisms that regulate BMAL1 upon clock synchronizer treatment, with a particular focus on the spatial distribution change of BMAL1 during early response towards synchronizers. We believe our result provides novel insights that BMAL1 Ser90 phosphorylation is essential for BMAL1 nuclear accumulation upon synchronizing factor stimulation, which could be one of the crucial mechanisms controlling cellular clock synchronization. Our findings also indicated that nuclear translocation of BMAL1 upon synchronizer treatment is circadian time-dependent, which further supports the idea that the BMAL1 cellular distribution change contributes to the clock resetting. In addition, we have now demonstrated that this phenomenon is observed across various biological contexts, including cell types (**Supplementary Figure 3**) and synchronization factors (**Supplementary Figure 4**), which suggests that this phenomenon may be conserved across peripheral clocks. To describe this, we added the following description to the introduction on page 3, lines 79-81;

‘However, there has been a lack of evidence concerning the mechanisms that regulate BMAL1, particularly regarding the subcellular localization change of BMAL1 during the early response to clock synchronizers.’

And modified a description on examination of BMAL1 nuclear localization upon Dex in pages 5-6, lines 129-147, with modifications underlined as,

‘Next, to investigate the commonality of the ISR among cell types, we used confocal imaging to examine whether the rapid accumulation of BMAL1 and CLOCK in the nucleus occurs after Dex stimulation in rodent cells with clock oscillation including rat glioma C6¹⁷, mouse myoblast C2C12¹⁸, and mouse fibroblast MEF cells¹⁴ (Supplementary Figure 3). C6 cells showed a slightly cytoplasmic enriched distribution of BMAL1, whereas BMAL1 localization was slightly nuclear for C2C12 and MEF cells. In all cells, BMAL1 nuclear translocation was observed, and CLOCK also tended to be enriched in the nucleus after the stimulation for MEF cells (Supplementary Figure 3b). These results suggest that BMAL1-ISR is preserved among diverse types of cells with a clock function. In addition, to test whether BMAL1-ISR occurs, we evaluated stimulation-mediated nuclear accumulation of BMAL1 with various clock synchronizers: mitogen (EGF), cAMP activator (Forskolin), calcium ionophore (A23187), and cellular stress (Heat-shock). In all cases, a homogenous accumulation of BMAL1 signal in the nuclei of NIH-3T3 cells was observed 20 min after each stimulation (Supplementary Figure 4a). The distribution of quantified nuclear-to-cytoplasmic ratio also supported that there was an increase in the ratio for BMAL1 upon all the stimulation tested (Supplementary Figure 4b, c). CLOCK accumulated uniformly in the nuclei 20 min after stimulation, as in the case of Dex stimulation, to a lesser extent compared to BMAL1. This result indicates that the BMAL1-ISR, a putative trigger of clock synchronization, was shown to be caused by various clock synchronizers.’

Revised Supplementary Figure 3.

Revised Supplementary Figure 4.

Furthermore, the data provided, particularly the live-cell imaging movies (I cannot see any described changes), fail to convincingly support the described phenomena.

Answer: We partly agree with the reviewer's comment that the nuclear-to-cytoplasmic localization change of the BMAL1 reporter is visually difficult to identify, which seems to be less pronounced compared to immunocytochemistry experiments. The discrepancy in visibility may be attributed to the fact that in the live-cell imaging assay, fusion of the reporter, mVenus, may have induced artifacts in the localization pattern of BMAL1. In addition, the expression of a single BMAL1 isoform is used, whereas in the case of IF, all the isoforms are considered. Despite these drawbacks, a major advantage of live-cell imaging is that it enables tracking changes in the same cell, which facilitates the analysis of local correlated behavior. In addition, we have now also examined the positive rate of ISR for given conditions, as shown in **Supplementary Figure 7c** to demonstrate its importance. Specifically, we evaluated the change in the nuclear-to-cytoplasmic ratio of each cell. We set a threshold for the ratio change based on the 75th percentile of the nuclear-to-cytoplasmic ratio change of the unstimulated condition. As a result, we found 74% BMAL1-ISR positive cells for EGF stimulation, and 80% for Dex stimulation (**Supplementary Figure 7c**). This result suggests that most of the cells stimulated with synchronization factors underwent BMAL1 nuclear accumulation. For the correlation index, we now calculated the time evolution of the index change, which suggested that the correlation index for synchronization factor stimulated cells increases over time and reaches a plateau at around 90 min post stimulation (**Supplementary Figure 7e**). These analyses suggests that after synchronization stimulation, the nuclear-to-cytoplasmic ratio among cells starts to correlate with each other to some extent.

We included the following description of the importance of live-cell imaging and the differences between IF and live-cell imaging on page 13, lines 409-420;

‘We used both immunocytochemical methods and live cell imaging with a monomeric fluorescent protein mutant mVenus fused to BMAL1 to observe changes in the nuclear and cytoplasmic distribution of BMAL1. While both methods resulted in the detection of immediate nuclear accumulation of BMAL1 as early as 15 minutes post clock-synchronizing stimuli, we detected slightly different localization patterns of BMAL1, with live cell imaging detecting less cytoplasmic localization of BMAL1. Although we used a *Bmal1* promoter-driven probe for the live imaging, there may be a potential difference in expression level that causes the localization difference. In addition, the possibility of artifacts from the fluorescent protein mVenus attached to BMAL1 with the linker should also be considered, as revealed in Supplementary Figure 6c. Nevertheless, both immunostaining and live cell imaging observed nuclear accumulation of BMAL1 upon clock synchronizer stimulation, suggesting that these results are not necessarily inconsistent with the BMAL1-ISR concept.’

And we described analysis of synchronization using live-cell imaging data on pages 7-8, lines 217-226, as follows;

‘To further confirm that these BMAL1 nuclear translocations are associated with clock synchronization, we introduced analytical approaches to assess the positive rate and correlation of BMAL1 localization changes among cells. First, to examine the percentage of cells that underwent nuclear translocation of BMAL1 after stimulation, we calculated the positive rate of BMAL1 nuclear localization by analyzing the change in the nuclear-to-cytoplasmic ratio of each cell. When taking the 75% quartile of the ratio for the unstimulated control as a threshold, the results showed that 74% of cells underwent nuclear translocation with EGF, and 80% with Dex stimulation, respectively (Supplementary Figure 7c). Next, we examined the correlation of BMAL1 localization changes among neighboring cells (local correlation index) and across the entire cell population (global correlation index) (Fig. 3e and Supplementary Figure 7d).’

And on page 8, lines 231-233 as follows;

‘We also evaluated the time evolution of these correlation indices and found that the correlation of localization increased after the stimulation (Supplementary Figure 7e).’

Revised Supplementary Figure 7c and 7e.

Besides, the study is confined to NIH-3T3 fibroblasts. While the manuscript claims BMAL1-ISR is a universal mechanism, it lacks validation in other cell types, tissues, or in vivo models to substantiate this claim.

Answer: We appreciate the reviewer’s concern about the universality of BMAL1-ISR, particularly regarding its specificity to NIH-3T3 cells. We have previously reported that BMAL1 is predominantly located in the nucleus in liver cells (Tamaru et al, 2003), and therefore localization analysis using tissue samples could be inconclusive. Instead, we examined BMAL1-ISR in various rodent cells that were confirmed to have clock oscillations, including rat glioma C6 cells (Koimuma et al, 2009), mouse fibroblast MEF cells (Tamaru et al, 2009), and mouse myoblast

C2C12 cells (Zhang et al, 2012), to investigate whether BMAL1-ISR could also be observed in these cells (**Supplementary Figure 3**). As a result, we found that BMAL1-ISR could be observed in all three cell types. In addition to showing that BMAL1 nuclear localization upon Dex stimulation occurs in various cell types, we also showed that this phenomenon is observed with different synchronizers in NIH-3T3 cells (**Supplementary Figure 4**). These findings raise the possibility that BMAL1-ISR could be a universal phenomenon underlying synchronization. However, we could not comprehensively analyze BMAL1-ISR, especially for tissues lacking a model cell line that exhibits circadian rhythmicity. Therefore, while we provide some evidence for BMAL1-ISR in various cells and stimulations, we rephrased the following discussion to address this concern, especially regarding the universality of this phenomenon.

We added and modified the description on the examination of BMAL1 nuclear localization upon Dex on pages 5-6, lines 129-147, with modifications underlined, as follows;

‘Next, to investigate the commonality of the ISR among cell types, we used confocal imaging to examine whether the rapid accumulation of BMAL1 and CLOCK in the nucleus occurs after Dex stimulation in rodent cells with clock oscillation including rat glioma C6¹⁷, mouse myoblast C2C12¹⁸, and mouse fibroblast MEF cells¹⁴ (Supplementary Figure 3). C6 cells showed a slightly cytoplasmic enriched distribution of BMAL1, whereas BMAL1 localization was slightly nuclear for C2C12 and MEF cells. In all cells, BMAL1 nuclear translocation was observed, and CLOCK also tended to be enriched in the nucleus after the stimulation for MEF cells (Supplementary Figure 3b). These results suggest that BMAL1-ISR is preserved among diverse types of cells with a clock function. In addition, to test whether BMAL1-ISR occurs, we evaluated stimulation-mediated nuclear accumulation of BMAL1 with various clock synchronizers: mitogen (EGF), cAMP activator (Forskolin), calcium ionophore (A23187), and cellular stress (Heat-shock). In all cases, a homogenous accumulation of BMAL1 signal in the nuclei of NIH-3T3 cells was observed 20 min after each stimulation (Supplementary Figure 4a). The distribution of quantified nuclear-to-cytoplasmic ratio also supported that there was an increase in the ratio for BMAL1 upon all the stimulation tested (Supplementary Figure 4b, c). CLOCK accumulated uniformly in the nuclei 20 min after stimulation, as in the case of Dex stimulation, to a lesser extent compared to BMAL1. This result indicates that the BMAL1-ISR, a putative trigger of clock synchronization, was shown to be caused by various clock synchronizers.’

In addition, we included a discussion on the universality of this phenomenon on page 12, line 368-377, with additions underlined, as follows;

‘Indeed, observation of nuclear BMAL1 accumulation suggested commonality of the BMAL1-ISR phenomenon across various biological contexts, including cell types and synchronization stimulation. However, since our focus was only on cell types and stimulation exhibiting circadian clock synchronization and oscillation, we could not comprehensively verify this phenomenon to validate its universality. Nevertheless, ISR could be one of the important molecular mechanisms governing synchronization, with an expectation of a new molecular target for inducing circadian clock synchronization. Further studies are warranted to determine whether

nuclear accumulation of BMAL1 or enhanced transcriptional activation capacity by nuclear BMAL1 during early clock synchrony is a universal feature of clock synchrony in different tissues and in vivo.’

And modified the description on the abstract on page 2, lines 28-30, with modification underlined as follows;

‘Here, we hypothesized that changes of the clock protein localization serve as a common synchronizing factor and investigated the relationship between BMAL1 and CLOCK localization pattern and clock synchronization in NIH-3T3 fibroblasts.’

References

1. Tamaru, T. et al. Nucleocytoplasmic shuttling and phosphorylation of BMAL1 are regulated by circadian clock in cultured fibroblasts. *Genes Cells* 8, 973–983 (2003).
2. Koinuma, S., Yagita, K., Fujioka, A., Takashima, N., Takumi, T. & Shigeyoshi, Y. The resetting of the circadian rhythm by Prostaglandin J₂ is distinctly phase-dependent. *FEBS Letters* **583**, 413–418 (2009).
3. Tamaru, T. et al. CK2 α phosphorylates BMAL1 to regulate the mammalian clock. *Nat. Struct. Mol. Biol.* 16, 446–448 (2009).
4. Zhang, X. *et al.* A non-canonical E-box within the MyoD core enhancer is necessary for circadian expression in skeletal muscle. *Nucleic Acids Res* **40**, 3419–3430 (2012).

Reviewer #2 (Remarks to the Author):

Tamaru and colleagues hypothesize that the nuclear accumulation of both BMAL1 and CLOCK after chemical synchronization in NIH 3T3 cells is a signal for molecular clock synchronization. The authors premise their hypothesis on a previously published observation of BMAL1 rhythms in cellular distribution after serum starvation in NIH 3T3 cells such that BMAL1 was predominantly found in the nucleus 4 hours after synchronization, more so in the cytoplasm after 16 hours, and again nuclear after 24 hours. While others have shown that PER1/2 expression increases as a result of various entrainment signals, including serum shock, the authors here posit that nuclear accumulation of the positive elements of the transcriptional feedback loop is a necessary predicating step for the PER1/2 ‘surge’. This hypothesis is intriguing for several reasons. If supported, this would introduce more than one novel target for modulating synchronization, which lies at the core of clock resetting; not just at the molecular level, but potentially for the whole organism.

Answer: We appreciate the reviewer for summarizing our manuscript with positive comments.

The authors tested their hypothesis using confocal imaging and subtractive imaging algorithms to compare the ratio of nuclear:cytoplasmic expression of BMAL1 and CLOCK after synchronization by various methods. While the approach is appropriate, the images provided are difficult to immediately interpret. I would recommend providing a merge of DAPI and BMAL1 / CLOCK signals to highlight the changes in nucleocytoplasmic ratios in the various preps. As it stands, the Western blots of separated nuclear and cytoplasmic fractions are more compelling and may be more appropriate as a main figure.

Answer: We thank the reviewer for the helpful suggestions. We now modified **Fig. 1a** to include a merged image of DAPI and BMAL1 signals, highlighting changes in nucleocytoplasmic ratios. In addition, we have placed the Western blots of separated nuclear and cytoplasmic fractions as the main figures in **Fig. 2d, e** (full-sized blot shown in **Supplementary figure 15**), with the quantification of nuclear BMAL1 abundance.

We also added the following description on the quantification of BMAL1 nuclear-to-cytoplasmic ratio on page 15, lines 470-475;

‘Briefly, the nuclei for each cell were segmented using the nuclear-stained image using a binarization program. For the cytoplasmic region, the boundary of the segmented nucleus image is expanded by 3 pixels by dilation. These segmented images were used as a mask image to obtain fluorescence intensity for each region by image calculation. Nucleus-to-cytoplasmic ratios were calculated by dividing the average nuclear fluorescence intensity by the average cytoplasmic intensity.’

Revised Fig. 1a.

Revised Fig. 2d, e.

The manuscript also investigates whether the endogenous clock affects BMAL1-based ISR and shows that there is a ‘time-of-day’ difference in the response. This does make sense, as the phase of molecular rhythms would be established by the initial synchronization signal, and it would be expected that nuclear BMAL1 would be lowest at around 12hrs post synchronization and highest after ~24 hours. I’m not sure this warrants a main figure to be honest and perhaps should trade places with the Western blots mentioned above.

Answer: We appreciate the reviewer for the comment. As the reviewer mentioned, we expected a ‘time-of-day’ difference in the BMAL1-based ISR, because BMAL1 shows changes in localization depending on the circadian phase. An additional important aspect of examining time-dependency was to investigate whether BMAL1-based ISR is interconnected with endogenous circadian oscillation, thereby supporting the idea that ISR contributes to clock synchronization. In this revision, we have also shown time-dependent nuclear enrichment of BMAL by Western blot. To provide more compelling evidence for BMAL1-based ISR, we have placed the Western blots in the main figure with a quantification of nuclear BMAL1 level (Fig. 2d, e) and instead replaced the histogram showing distribution of BMAL1 location on the supplementary figure (Supplementary Figure 5b).

BMAL1 nuclear accumulation was further confirmed in single cells via timelapse imaging of Bmal1::luc reporter in BMAL1-deficient cells. The authors highlight cells that undergo ‘ISR response’, but this does not seem universal across all cells, which begs the question of how important this is if the ensemble signal is rhythmic (and therefore synchronized). Can the authors quantify how many cells were positive for ISR, and what does this mean in terms of global synchronization?

Answer: We appreciate the constructive suggestions on the analysis. The correlation index shown in Fig. 3f is based on the correlation of nuclear-to-cytoplasmic ratio changes over time among cells and is therefore not a direct measure of the positive rate for BMAL1-ISR. To address this, we evaluated the nuclear-to-cytoplasmic ratio change of each individual cell. We set a threshold for the ratio change based on the 75th percentile of the change observed under unstimulated conditions. As a result, we found that 74% of cells were BMAL1-ISR positive upon EGF stimulation, and 80% upon Dex stimulation (**Supplementary Figure 7c**). These results suggest that majority of the cells stimulated with synchronization factors underwent BMAL1 nuclear accumulation.

For the correlation index, we have calculated the time evolution of the index change, which revealed that the correlation index for synchronization factor-stimulated cells increased over time and reached a plateau at around 90 min post-stimulation (Supplementary figure 6e). This analysis suggests that after synchronization stimulation, the nuclear-to-cytoplasmic ratio among cells becomes correlated to some extent. Please note that in the analysis of the correlation index, all the data from the start of observation to a given time were used, so that the period when the cell's nuclear-to-cytoplasmic ratio was not correlated was included, which could have led to a smaller percentage of correlated cells in terms of global synchronization. To describe this, we added the following description on pages 7-8, lines 217-226;

'To further confirm that these BMAL1 nuclear translocations are associated with clock synchronization, we introduced analytical approaches to assess the positive rate and correlation of BMAL1 localization changes among cells. First, to examine the percentage of cells that underwent nuclear translocation of BMAL1 after stimulation, we calculated the positive rate of BMAL1 nuclear localization by analyzing the change in the nuclear-to-cytoplasmic ratio of each cell. When taking the 75% quartile of the ratio for the unstimulated control as a threshold, the results showed that 74% of cells underwent nuclear translocation with EGF, and 80% with Dex stimulation, respectively (Supplementary Figure 7c). Next, we examined the correlation of BMAL1 localization changes among neighboring cells (local correlation index) and across the entire cell population (global correlation index) (Fig. 3e and Supplementary Figure 7d).'

And on page 8, lines 231-233 as follows;

'We also evaluated the time evolution of these correlation indices and found that the correlation of localization increased after the stimulation (Supplementary Figure 7e).'

Revised Supplementary Figure 7c and 7e.

The authors hypothesize that phosphorylation of Ser90 by CK2 – a prior discovery by the authors – is crucial to BMAL1 nuclear entry in ISR. They show immediate increase in Ser90 phosphorylation after synchronization, which is inhibited by a CK2-specific inhibitor. Importantly, they show that CK2 phosphorylation is needed for the Dex-induced phase shifts in Per2 and Bmal1 that allow for synchronization.

Answer: We thank the reviewer for judiciously summarizing our manuscript.

Finally, the authors conducted in silico experiments based on the Kim-Forger computational model. While the results of these experiments all seem to align with the cell-models, the utility of the model is uncertain, mainly due to several parameters of the model being ‘arbitrary’ or, as stated by the authors, adjusting the BMAL1 phosphorylation rate constant coefficient by a factor of 1 – 50. There may be interest in how this value affects the simulation; is the rate of BMAL1 phosphorylation always constant? The authors should provide more comprehensive data, including the values that did not align with their results, and provide an explanation in terms of the mechanism they propose for why their specified values ‘worked’.

Answer: We appreciate the reviewer’s comment on the simulation analysis. In our simulation, we additionally included the factor “phosphorylated BMAL1” in the original model. Our simulation results confirmed that inclusion of phosphorylated BMAL1 did not perturb the core clock’s oscillation, justifying this inclusion (Supplementary Figure 9b). Then, we modified the values of two additional parameters to this model, the duration and magnitude of the increase in BMAL1 phosphorylation rate. Although the values for these parameters were arbitrarily determined to match the phosphorylation level change obtained from the Western blotting experiment, we

investigated how the duration and magnitude of phosphorylation increase impact the maximum level of Ser90 phosphorylation induction, *Per2* expression induction, and phase shift of *Per2* oscillation (**Supplementary Figures 10 and 11**). As expected, longer duration and higher magnitude of phosphorylation rate change tended to induce larger effects on these three parameters. Notably, for a given rate change, the maximum Ser90 phosphorylation level remained consistent regardless of the duration except for a very short period, 0.1 h (**Supplementary Figure 10b**). We also investigated the effect of differences in temporal patterns of phosphorylation rate increase. To examine whether the pattern of rate increase affects the simulation, we performed a simulation with two distinct temporal patterns, one in which the rate increase is constant, and the other in which the rate increase is in a pulse-like pattern (**Supplementary Figure 10c**). The results suggested that the temporal pattern does not affect the simulation except for the maximum level of Ser90 phosphorylation induction which can be explained by the fact that the maximum level is proportional to the rate increase, which was set higher in the pulse-like temporal pattern to adjust the net rate increase over the synchronization duration. Accordingly, we decided to use a constant rate of increase to simplify the simulation.

Based on these findings, we chose a 15-fold increase in the BMAL1 phosphorylation rate for synchronization, as this factor closely mirrored the experimentally observed fold change. Since we applied synchronizer treatment for a time span of 2 hours in our experiments, we set the duration of the rate increase to match this timeframe. We noticed that for a factor of greater than 20 for the phosphorylation rate of BMAL1, and a duration exceeding 4 hours could introduce a time-dependent phase shift in *Per2* that might confound our simulation outcomes (**Supplementary Figure 11a**). In these conditions, although the time-dependency trend was similar for *Per2* induction, the magnitude of phosphorylation rate increase does not correlate with the induction level of *Per2*. Possibly, correlational relationships are not maintained due to complex factors such as the time-dependent expression of circadian genes and phase shifts induced by the rate change of phosphorylation. Nevertheless, with our selected factor of 15, the duration of the increase does not induce this particular phase shift, solidifying our parameter selection. Thus, we concluded that a 2-hour duration and a 15-fold increase in the BMAL1 phosphorylation rate were the optimal parameters for simulating BMAL1-based ISR.

To describe these, we added the following description on quantification of BMAL1 nucleus-to-cytoplasmic ratio on pages 9-10, lines 299-313 as follows;

‘In this simulation, the model had two arbitrarily defined values: the magnitude of the phosphorylation rate constant change and duration (Supplementary Figure 10a). Hence, we first simulated the impact of the level of rate change and duration on the maximum level of Ser90 phosphorylation induction (Supplementary figures 10b and 11). We found that for a given rate change, the maximum Ser90 phosphorylation level remained consistent regardless of the duration,

except for a duration of 0.1 h. Considering the experimentally obtained BMAL1 Ser90 phosphorylation induction fold change (Figure 4b), we set the magnitude to a 15-fold increase. We also confirmed that a simple, constant rate increase produced nearly identical outcomes to a complex pulse-like pattern, so we adopted the constant rate for simplicity (Supplementary Figure 10c). We note that although longer duration and higher levels of phosphorylation rate change tended to induce a larger induction level of *Per2*, as well as a phase shift of *Per2* oscillation (Supplementary Figure 11), duration had a limited influence on the predetermined magnitude of a 15-fold increase in the phosphorylation rate. Consequently, the duration was set to two hours, which corresponds to the duration of the synchronization experiment.’

And on page 10, lines 321-323 with modifications underlined as follows;

‘Our simulations revealed a pronounced circadian time-dependency in the magnitude of the *Per2* phase shift (Fig. 5b and Supplementary Figure 11a).’

We also revised our description of procedure for the simulation in pages 18-19, lines 600-612 with modifications underlined as follows;

‘For the simulation of the effect of BMAL1-Ser90 phosphorylation on *Per2* expression, we first evaluated the optimal coefficient for the rate constant for BMAL1 phosphorylation (Phos rate) by a factor ranging from 1 to 50, and also the duration of Phos rate increase ranging from 0.1 h to 8 h. The range for Phos rate was determined based on the report that activation change of the potential responsible kinase for BMAL1 Ser90, CK2, would occur in the order of 10^{39} , and that positive effect would be on the Phos rate. The range for duration was based on the assumption that an increase in Phos rate would occur transiently, but could range from short to long duration. We systematically applied a combination of Phos rate increase (1, 2, 3, 5, 10, 15, 20, 30, and 50) and duration (0.1, 0.2, 0.5, 1, 2, 4, 6, and 8 hours) at every 1 hour between simulated circadian time of 8-32 hours as a simulated synchronization effect. Fold change of BMAL1 phosphorylation and *Per2* expression was defined as the ratio of maximum induction levels after simulated synchronization, and *Per2* phase shift was calculated as the phase shift of *Per2* oscillation compared to simulation without any Phos rate change.’

Revised Supplementary Figure 10.

Revised Supplementary Figure 11.

Revised Supplementary Figure 12b.

Overall, the manuscript is presented well, there are no glaring grammatical issues, and the methods are sensibly described. The simulation methods should be more clearly described when referring to the input values. The discussion of the results is in-line with the results and notes the limitations of the study.

Answer: We appreciate the reviewer again for the constructive comments and suggestions on our manuscript.

Reviewer #3 (Remarks to the Author):

This paper investigates how circadian clocks synchronize at the cellular level. The authors focus on the localization patterns of clock proteins BMAL1 and CLOCK in NIH-3T3 fibroblasts. They show that synchronized nuclear accumulation of BMAL1 and CLOCK occurs immediately after clock-resetting treatments (ISR). The study further shows that phosphorylation of BMAL1 at Ser90 by CK2, which promotes its nuclear accumulation, is also rapidly increased during this response. Inhibiting CK2 partially suppressed the acute increase of Per2 and disrupted clock resetting. Computational simulations supported the idea that increased BMAL1 phosphorylation and nuclear localization can reset the circadian clock.

1. The data presented in Figure 1 is not very convincing. The representative immunohistochemical images don't show much difference in signal between the different conditions. However, it seems that the unstimulated WT condition has significantly more cells compared to the other conditions, which may indicate that Dex caused cell death. If this is not the case, please choose better images that more accurately reflect the analyzed data.>

Answer: We appreciate the reviewer for the critical comments on the immunohistochemical images. Indeed, the current representative data is somewhat less convincing as the apparent nucleus and cytosol are difficult to distinguish due to a large field of view. To address this, we have now enlarged the view to facilitate visibility of BMAL1 localization within a cell. In addition, we agree that the presented data has a difference in cell density, which could raise concerns about artifacts regarding cell viability. To improve this, we have now used replicates with similar cell density to other panels (**Fig. 1a**) and placed the original magnification image on **Supplementary Figure 14a**. Please note that Dex stimulation could induce variation in cell density across the dish; however, its effect on cell viability is negligible.

Revised Fig. 1a.

2. Figure 1B – It seems that the Clock Nuc/Cyt ratio changes in response to longer treatment with Dex, while BMAL1 does not exhibit a similar time-dependent effect. Please explain. >

Answer: We appreciate the reviewer's comment. It has been reported that nuclear/cytoplasm distribution of CLOCK was found to be under BMAL1 control (Kwon et al 2006, Kondratov et al, 2003), suggesting that CLOCK accumulation in the nucleus is dependent on BMAL1. Therefore, BMAL1 accumulation may precede CLOCK. Indeed, our immunocytochemical analysis showed that the correlation of the nuclear-to-cytoplasmic ratio of BMAL1 and CLOCK increases upon stimulation and takes the highest positive correlation at 40 min post stimulation (Fig. 1c), supporting that CLOCK nuclear localization is dependent and follows BMAL1 nuclear localization. To describe this, we added the following description on page 5, lines 124-126 as follows;

'Since CLOCK nuclear localization is dependent on BMAL1^{8,16}, increase in the correlation over time suggest clock proteins translocate to the nucleus sequentially in the order of BMAL1 and CLOCK.'

References

1. Kwon, I. et al. BMAL1 Shuttling Controls Transactivation and Degradation of the CLOCK/BMAL1 Heterodimer. *Mol. Cell. Biol.* 26, 7318–7330 (2006).
2. Kondratov, R. V. et al. BMAL1-dependent circadian oscillation of nuclear CLOCK: posttranslational events induced by dimerization of transcriptional activators of the mammalian clock system. *Genes Dev.* 17, 1921–1932 (2003).

3. Similarly to comment 1, there seems to be no change in signal between stimulated and unstimulated cells, which does not align with the analysis on the right side of Figure 2a. Please address this accordingly.

Answer: We appreciate the reviewer's comment. For the two conditions displayed in Fig. 2a, they are representative images showing the result for Dex 18h and Dex 22h samples, while the quantification results on the right side (the original Fig. 2b, now on Supplementary Figure 4b) are histograms for Dex 15h, Dex 18h, and Dex 22h samples. We apologize if this was confusing. Please note that we have replaced this panel with the Western blot result after fractionation.

As for the results, the representative panel for Dex 18h samples shows that BMAL1 is distributed in both the nucleus and cytoplasm and undergoes nuclear translocation upon Dex stimulation. In contrast, the representative panel for Dex 22h samples shows that BMAL1 primarily locates in the nucleus before the stimulation and does not substantially change its location upon Dex stimulation. Accordingly, the quantification results in Fig. 2b suggest that there is nuclear translocation of BMAL1 on the Dex 18h sample but not on the Dex 22h sample (second row and third row of the original Fig. 2b). Therefore, we believe the representative figure aligns with the quantified result. Please note that the quantification suggested that the nucleus-to-cytoplasmic ratio is around 2 for the Dex-stimulated-Dex 18h sample; therefore, some population of BMAL1 would also be located in the cytoplasm even with the stimulated condition.

4. Supplementary Figure 3 – There is no evident change in signal between the conditions in the representative images, which would indicate immediate nuclear BMAL1 accumulation compared to control.

Answer: We appreciate the reviewer's comment on the issue of visibility of nuclear accumulation. We have now improved visualization by enlarging each panel to enable distinguishing the nucleus and the cytoplasmic region of the cell (**Supplementary Figure 4a**, and the uncropped image of the original one on **Supplementary Figure 14b**). In addition, we quantified the cellular BMAL1 distribution to ensure that there is a nuclear enrichment of BMAL1 after each stimulation (**Supplementary Figure 4b, c**). To describe these, we included the following description on quantification of BMAL1 nucleus-to-cytoplasmic ratio on page 6, lines 142-144 as follows;

‘The distribution of quantified nuclear-to-cytoplasmic ratio also supported that there was an increase in the ratio for BMAL1 upon all the stimulation tested (Supplementary Figure 4b, c).’

5. Figure 4 – For better accuracy, BMAL1 should be isolated from the nuclear fraction rather than cellular BMAL1. Hence, the housekeeping gene should be a nuclear marker and not actin.

Answer: We thank the reviewer for the comment. We partly agree that the BMAL1 abundance change in the nuclear fraction would better represent the phosphorylation level change of BMAL1 on Ser90 in the nucleus. Indeed, fractionation analysis supported the increase of phosphorylated BMAL1 in the nucleus (**Supplementary Figure 8c**). However, we considered that Ser90 phosphorylation also occurs in the cytoplasm, which was also demonstrated in our previous research (Tamaru et al, 2015 Plos Biol.). Therefore, we believe cellular BMAL1 better represents our experimental model, a rate increase in the BMAL1-Ser90 phosphorylation, followed by nuclear enrichment of BMAL1.

To describe this, we added the explanation on quantification of nuclear BMAL1 Ser90 phosphorylation on page 8, lines 249-252 as follows;

‘Moreover, the BMAL1-Ser90 phosphorylation signal was also increased in nuclear fractionated samples by Dex stimulation but suppressed by GO289 stimulation, supporting that CK2 inhibition suppresses nuclear BMAL1 phosphorylation at Ser90 (Supplementary Figure 8c).’

Revised Supplementary Figure 4.

Revised Supplementary Figure 8c.